# Bioinformatic Analysis of Secondary Metabolite Biosynthetic Potential in Pathogenic *Fusarium*

**DOI:** 10.3390/jof9080850

**Published:** 2023-08-15

**Authors:** Chao Lin, Xi-long Feng, Yu Liu, Zhao-chen Li, Xiu-Zhang Li, Jianzhao Qi

**Affiliations:** 1Shaanxi Key Laboratory of Natural Products & Chemical Biology, College of Chemistry & Pharmacy, Northwest A&F University, Yangling, Xianyang 712100, China; 2State Key Laboratory of Plateau Ecology and Agriculture, Qinghai Academy of Animal and Veterinary Sciences, Qinghai University, Xining 810016, China

**Keywords:** pathogenic *Fusarium*, fungal toxin, biosynthesis, clustering analysis

## Abstract

*Fusarium* species are among the filamentous fungi with the most pronounced impact on agricultural production and human health. The mycotoxins produced by pathogenic *Fusarium* not only attack various plants including crops, causing various plant diseases that lead to reduced yields and even death, but also penetrate into the food chain of humans and animals to cause food poisoning and consequent health hazards. Although sporadic studies have revealed some of the biosynthetic pathways of *Fusarium* toxins, they are insufficient to satisfy the need for a comprehensive understanding of *Fusarium* toxin production. In this study, we focused on 35 serious pathogenic *Fusarium* species with available genomes and systematically analyzed the ubiquity of the distribution of identified *Fusarium*- and non-*Fusarium*-derived fungal toxin biosynthesis gene clusters (BGCs) in these species through the mining of core genes and the comparative analysis of corresponding BGCs. Additionally, novel sesterterpene synthases and PKS_NRPS clusters were discovered and analyzed. This work is the first to systematically analyze the distribution of related mycotoxin biosynthesis in pathogenic *Fusarium* species. These findings enhance the knowledge of mycotoxin production and provide a theoretical grounding for the prevention of fungal toxin production using biotechnological approaches.

## 1. Introduction

The genus *Fusarium* is a widespread and diverse group of filamentous fungi found in soils worldwide and interacting with various plants [1,2]. Although the majority of *Fusarium* species are harmless soil microorganisms, a small number of pathogenic *Fusarium* spp. pose a significant threat to agriculture, the food industry, and human and animal health [1,3,4,5]. These pathogenic species are capable of infecting a wide range of plants, including food and cash crops, medicinal plants and ornamentals [6,7,8]. Infection caused by *Fusarium* leads to different types of rot, such as root, stem, basal, flower and spike rot, by attacking and destroying the plant’s vascular system [1,9,10]. In addition, *Fusarium* also damages plants by producing its own toxins, resulting in wilting, plant death and ultimately reduced crop yield or ornamental value of landscape plants [1,3,11]. The control of pathogenic *Fusarium* species is therefore a challenging task in agricultural production.

Mycotoxins, toxic secondary metabolites produced by filamentous fungi, are a threat to human and animal health, and *Fusarium* is a major producer of these toxins [10]. Almost all *Fusarium* species produce these harmful chemicals. For example, *F. fujikuroi*, a globally distributed pathogen, causes Bakanae disease in rice by producing gibberellin, which causes abnormal stem elongation in the host plant [12]. Several studies have shown that *Fusarium* is capable of producing structurally complex toxic metabolites such as terpenes (e.g., T-2 toxin [13], deoxynivalenol (DON) [14], fusarenone X [15] and gibberellin [16]), polyketides (PKs, e.g., fumonisin B1 [17], fusaric acid [18] and zearalenone [19]), nonribosomal peptides (NRPs, e.g., enniatin A1 [20], beauvericin [21] and apicidin [22]) and hybrid compounds (e.g., fusarin C [23], equisetin [24] and (-)-sambutoxin [25]). These toxins not only contribute to the symptoms of plant infections but also enter the food chain and affect human and animal health, in some cases causing serious damage or even loss of life.

The identification of core enzymes and gene clusters is of great importance in biosynthetic studies. In the case of well-documented compounds such as terpenoids, polyketides and nonribosomal peptides, the core enzyme plays a crucial role in determining the structural basis of the compound. On the other hand, gene clusters are responsible for the diversity and complexity of these compounds. To aid these studies, Synthaser software [26] not only provides a visual representation of the domain composition of multidomain enzymes but also allows the simultaneous alignment of multiple enzymes. Furthermore, Clinker software [27] allows a comprehensive comparison of the correspondence between biosynthetic gene clusters (BGCs) and the genes within them on a large scale. Finally, Big-Scape software [28] takes the understanding of correlations and similarities between biosynthetic gene clusters to an advanced level. These software packages for biosynthesis research offer immense convenience and utility.

Although previous studies have discussed *Fusarium*-derived mycotoxins and provided insights into the biosynthetic pathways of specific toxic compounds, little attention has been paid to exploring the prevalence and variability of BGCs for these toxins within *Fusarium* species. This study aims to address this gap by applying bioinformatics-based genome-mining techniques to 35 pathogenic *Fusarium* species, which provide valuable genomic information related to natural product biosynthesis. Through our comprehensive bioinformatic analysis, we have successfully identified more than 30 classes of enzymes involved in terpene synthesis, as well as more than twenty classes of nonribosomal peptide synthetases (NRPSs) and polyketide synthases (PKSs), and more than ten classes of PKS-NRPS hybrids. These findings open new avenues for future biosynthetic investigations and provide a theoretical basis for using biotechnology to mitigate the harmful effects caused by these pathogenic fungi.

## 2. Materials and Methods

### 2.1. Strains and Genome Sequences

A comprehensive collection of 35 pathogenic *Fusarium* species was used in this study, 34 of which were obtained from the NCBI Genbank and one from the JGI genome portal. The specific details of these pathogenic *Fusarium* species can be found in Appendix A.

### 2.2. Gene Cluster Prediction and Similarity Network Analysis

The prediction of BGCs within the genomes of the 35 pathogenic *Fusarium* species was performed using the online tool antiSMASH 7.0 [29]. The input files consisted of the genome sequence in fasta format and the corresponding annotation file in gff3 format. The run parameters were set as follows: detection stringency was relaxed; all extra features were enabled; and time-consuming features were enabled. BGCs were categorized into seven types, including NRPS (including NRPS-like), PKS, hybrid (PKS-NRPS), RIPP, terpene, indole and others. The “others” category included rare BGC types such as phosphonate, NI-siderophore, cyclic dipeptide synthase (CDPS) and phosphonate-like.

A similarity network of the BGCs among the 35 different *Fusarium* genomes was constructed using BiG-SCAPE v1.1.5 [28]. The following parameters were used: bigscape.py -i input -o output —cutoffs 0.5 —mibig21. Each node within the network represents a distinct BGC, and those with similar Pfam domain metrics were connected by edges. A cutoff of 0.5 was used for analysis, and the resulting similarity network was visualized using Cytoscape 3.0.9 (https://cytoscape.org).

### 2.3. Cluster Analysis Based on Evolutionary Trees and Sequence Identity Analysis

A clustering analysis of all enzymes possessing a specific function was performed by constructing an evolutionary tree. A maximum likelihood phylogenetic tree was constructed using IQ-TREE v. 2.2.0 [30]. The best-fit model was determined using the model-finding method [-m M -nt AUTO], followed by the construction of the evolutionary tree using the identified best-fit model [-m “best-fit model” -bb 1000 -alrt 1000 -abayes -nt AUTO]. The resulting tree was visualized and annotated using the website iTOL V6 [31].

The identified enzymes from *Fusarium* species were used as cues for cluster analysis. Clusters that fell within the same branch as the enzymes and had an identity of at least 50% were designated as independent groups. Identity assessment was performed using the percent identity matrix of the website Cluster Omega (https://www.ebi.ac.uk/Tools/msa/clustalo/, accessed on 5 July 2023).

### 2.4. Structural Analysis of Multidomain-Containing Enzymes

For a detailed analysis of PKS, NRPS or NRPS-like enzymes and PKS-NRPS and NRPS-PKS hybrid synthases, the tool synthaser [26] was used to analyze their domain characteristics. This analysis included different domains such as adenylation (A), acyl carrier protein (ACP), ACP synthase (ACPS), acyl transferase (AT), peptidyl carrier protein (PCP)/thiolation (T), thioesterase (TE), condensation (C), dehydrogenase (DH), epimerization (E), enoyl reductase (ER), ketoreductase (KR), beta-ketoacyl synthase (KS), methyltransferase (MT), product template (PT), starter unit: acyl carrier protein transacylase (SAT), thioesterase (TE), thioester reductase (TR) and carnityl acyltransferase (AT).

### 2.5. Homology and Similarity Analysis of BGCs

To identify published *Fusarium*-derived BGCs, the UniProtKB/Swiss-Prot database was subjected to Blastp analysis using the predicted core enzymes of *Fusarium* BGCs. The best-matching BGCs were then extracted from public databases such as NCBI and MIBiG [32]. In cases where a fully annotated BGC was available but not deposited in a public database, the relevant information was obtained directly from the corresponding genome sequence.

The assessment of homology and similarity between two or more BGCs was performed using Clinker, which is based on the comparison of the sequence similarity of the encoded proteins. Visualization of the comparison results was achieved using clustermap.js [27], a tool embedded in Clinker to generate gene cluster comparison plots.

### 2.6. Three-Dimensional Structure Modeling and Prediction of Proteins

The three-dimensional structure modeling and prediction of proteins were performed through a locally deployed AlphaFold database [33], and the AlphaFold DB version is V2.3.0.

## 3. Results

### 3.1. Biosynthetic Classes and Network Analysis for the BGCs from Pathogenic Fusarium Species

To maximize the identification of BGCs in the genomes of selected *Fusarium* species, the annotation information of 35 pathogenic *Fusarium* genomes was analyzed for prediction using the antiSMASH tool. A total of 1733 putative BGCs were detected, with an average of 51 BGCs per species. Among these species, *F. haemophilus* had the highest number of BGCs (65), whereas *F. kerogenes* and *F. bacilli* had the lowest number of BGCs (38) (Figure 1, Appendix A). To investigate whether the number of BGCs present in different species correlated with their evolutionary relationships, an evolutionary tree based on single-copy orthologous genes was constructed, which revealed two distinct branches containing these species. Six species, including *F. decemcellulare*, formed a smaller branch, while the remaining twenty-nine species were grouped in the other branch. However, there was no clear differentiation between the branches, nor was there any uniformity within them (Figure 1A). In terms of BGC types, each species had the largest number of NRPS BGCs, and most species contained dimethylallyltryptophan synthase (DMATS) labeled as indole, although the number of these DMATs did not exceed three (Figure 1B).

To gain a deeper understanding of these BGCs, a gene cluster family (GCF) network analysis was performed using BiG-SCAPE. Unfortunately, due to compatibility issues between BiG-SCAPE V1.15 and antiSMASH V7.0.0, a total of 112 RiPP-type BGCs and 26 other-type BGCs (NRP-metallophore and phosphonate) could not be identified and classified by the BiG-SCAPE pipeline. Nevertheless, BiG-SCAPE successfully classified 1640 *Fusarium*-derived BGCs and 1918 identified BGCs into 141 GCFs and 2047 individual clusters (Appendix A) based on the similarity of predicted protein-coding domains. Within the GCF networks consisting of more than ten BGCs, several networks were observed that consisted exclusively of a single type of BGC. Specifically, twelve networks consisted exclusively of NRPS BGCs, nine networks consisted exclusively of terpene BGCs, five networks consisted exclusively of type I polyketide synthase (PKS) BGCs, and one network consisted of PKS_other BGCs (Figure 2). Among the 141 GCF networks, the most complex mixed network was formed by 70 PKS_NRPS hybrid BGCs, 16 type I PKS BGCs and 2 NRPS BGCs, giving a total of 19 identified hybrid BGCs. Based on these results, seventeen GCF networks were identified, including five type I PKS GCF networks (BIK_GCF, alt_GCF, DEP_BGC, ACTT/PKS19_GCF and fsr_GCF), four NRPS GCF networks (APS_GCF, aba_GCF, san_GCF and chry_GCF), four terpene GCF networks (Ffsc4_GCF, SQS1_GCF, GA_GCF and tri_GCF), two PKS_NRPS hybrid GCF networks (ZEA_GCF and FSL_BGC) and two other GCF networks (has_GCF and fsd_GCF) (Figure 2, Appendix A).

### 3.2. Terpene Biosynthetic Pathway of Pathogenic Fusarium Species

A total of 430 fundamental genes involved in terpene production were identified from a set of 1733 BGCs using antiSMASH. These genes encode enzymes such as sesquiterpene synthase, geranylgeranyl pyrophosphate (GGPP) cyclase, sesterterpene synthase, triterpene synthase, lycopene cyclase/phytoene synthase, as well as GGPP synthase for the production of diterpene scaffolds and the conventional pentenyltransferases (PTs) and indole moiety-specific dimethylallyltryptophan synthase (DMATS). Among these genes, the largest number is associated with sesquiterpene synthases, accounting for almost half of the total, while the least number of genes belong to the sesterterpene synthase group, with only twelve sequences (Appendix A).

A phylogenetic tree was constructed using 209 sesquiterpene synthases, and the results showed a strong clustering pattern. Among these clusters, seven identified sesquiterpene synthases provided strong evidence for the identification of related sequences. Ffsc4, a multiproduct sesquiterpene cyclase in the pathogenic fungus *F. fujikuroi*, produces not only the 4/9 bicyclic 2-*epi*-(*E*)-β-caryophyllene and (*E*)-β-caryophyllene but also the 11-membered α-humulene [34]. Ffsc4 has 29 homologous sequences in the 35 selected pathogenic *Fusarium* species, and their sequence identities are higher than 69% (Appendix A). Therefore, it is speculated that the products of this cluster are identical or similar to these three sesquiterpenes. Ffsc6 is another multiproduct sesquiterpene cyclase from *F. fujikuroi* which produces *α*- and *β*-cedrene, *α*-acoradiene, *α*-alaskene and *β*-bisabolene, along with other sesquiterpenes [34]. Ffsc6 has seventeen homologous sequences among the selected pathogenic *Fusarium* species, and their sequence identities are higher than 59% (Appendix A). It is speculated that the products of this cluster are the same or similar to these five sesquiterpenes. STC5 and STC3 are the other two sesquiterpene cyclases from *F. fujikuroi* [35]. The former has 19 sequences with high sequence identities (>78%, Appendix A), and their products are probably the same as guaia-6,10(14)-diene, a 5/7 bicyclic sesquiterpene synthesized by STC5 [35]. The latter, whose product is the bicyclic sesquiterpene eremophilene (A10) [35], has two homologous sequences with identities as high as 81% (Appendix A).

CML1 is a sesquiterpene alcohol synthase found in *F. graminearum*, a pathogenic fungus affecting cereal crops [36]. CML1 is responsible for the biosynthesis of longiborneol [37]. CML1 and six sesquiterpene synthases from the pathogenic *Fusarium* species form a cluster with more than 71% sequence identity (Appendix A). It is speculated that the products of this cluster are identical or similar to longiborneol. Trichodiene synthase, also known as TOX5 (TRI5), was initially identified in the pathogenic fungi *Gibberella pulicaris* (*F. sambucinum*) [38] and *F. sporotrichioides* [39] and subsequently in *F. pseudograminearum* [40] and *F. graminearum* [41]. Members of the TRI5-containing cluster share more than 85% sequence identity (Figure 3, Appendix A), and these putative BGCs contain TRI5 homologous sequences with high similarity (Figure 4A). Trichodiene serves as the basic structure for various fungal sesquiterpene toxins, including DON, nivalenol (NIV) and T-2 toxins (Figure 4B), which are produced by *Fusarium* and *Stachybotrys* species [42]. FlvE, a terpene cyclase responsible for the synthesis of (1*R*,4*R*,5*S*)- (+) -acoradiene in *Aspergillus flavus*, has five homologues with more than 47% sequence identity in the selected *Fusarium* species (Appendix A). Further comparison of the BGCs revealed that four genes in the *flv*BGC share similarities with genes from *Fusarium*-derived BGCs (Appendix A). 

In filamentous fungi, the synthesis of GGPP and its subsequent cyclization is carried out by two different enzymes which work together to produce the diterpene backbone. In these selected *Fusarium* species, the 87 enzymes involved in the synthesis of the diterpenoid skeleton have been classified into three groups, consisting of 40 GGPP synthases, 20 GGPP cyclases and 27 GGPP-related cyclases, respectively (Appendix A). The gene *dpfgD*, which is involved in the biosynthesis of the diterpenoid pyrone subglutinol A [43], together with the three GGPP synthases, forms a smaller subgroup within the GGPP synthase group with an identity of over 86% (Appendix A). On the other hand, the gene GGS [44], together with the remaining 34 GGPP synthases, forms a larger subgroup with more than 81% identity (Appendix A). The gene CPS/KS [16,45], an *ent*-kaurene synthase identified in *F. fujikuroi*, plays a key role in the biosynthesis of gibberellin. The sequence identity between CP/SKS and the other members of the GGPP synthase group is over 41% (Appendix A). The terpene cyclase gene, *dpfgB*, is responsible for the cyclization of oxidized furanone diterpene [43], and its 26 homologues have been identified in the selected *Fusarium* species, all of which share more than 57% sequence identity (Appendix A). Two diterpenoid synthesis gene clusters, GABGC and *dpfg*BGC, have been discovered in *Fusarium* and have served as lead examples in the search for several similar BGCs in the selected 35 *Fusarium* species (Appendix A). 

Sesterterpenoids are a minority within the terpene family, and filamentous fungi are their primary producers. In *Fusarium* species, these compounds are mainly derived from their pathogenic members. Fusaproliferin is a toxic compound found in the eggplant disease-causing pathogen *F. solani* [46]. Mangicdiene and variecoltetraene are products catalyzed by FgMS from *F. graminearum* [47,48], while fusoxypenes A–C are products catalyzed by FgMS from *F. oxysporum* [49] (Figure 5A). Six sesterterpene synthases were identified in the 35 *Fusarium* species, including two chimeric enzymes from FgMS in *F. graminearum* [47,48] and FoFS in *F. oxysporum* [49]. The predicted three-dimensional structure based on Alphafold revealed that the four sesquiterpene synthases, as well as the two identified chimeric enzymes, possessed two relatively independent functional domains (Appendix A). Cluster analysis of these twelve sequences divided them into three clades (Figure 5B). The three uncharacterized sesquiterpene synthases are in an independent clade with up to 79% sequence identity among the three, while the amino acid sequence identity between these three and the two characterized sesquiterpene synthases is no more than 30% (Figure 5C, Appendix A). FgMS and FGSG_01738 not only show high identity in the primary sequences (Figure 5C) but also show highly similarity with a 1.012 Å *RMSD* in the three-dimensional structures (Figure 5D). Spatial comparisons also revealed the similarity of the three uncharacterized sesterterpene synthases, with RMASD values ranging from 0.366 Å to 2.259 Å (Figure 5E).

Epoxysqualene cyclase utilizes 2,3(*S*)-epoxysqualene as a substrate to synthesize the triterpene backbone with diverse structural characteristics. Among the selected *Fusarium* species, ten lanosterol synthases and one squalene hopane cyclase (SHC), FDECE_14603, were identified (Appendix A). The ten putative lanosterol synthases shared a significant identity (>49%) with Erg7 from *F. graminearum* [50] (Appendix A). FDECE_14603 displays 55% sequence identity with the identified SHC, Aafum, from the human pathogen *A. fumigatus* [51]. Carotenoids are a group of natural terpenoid pigments with a C40 backbone abundant in filamentous fungi. In *Fusarium fujikuroi*, the complete carotenoid biosynthesis pathway has been elucidated (Figure 6), with *carRA* being the crucial gene responsible for the synthesis of the C40 backbone [52,53]. Using *carRA* as a reference, seventeen homologous genes were screened, and these genes exhibited 94% sequence identity (Appendix A). Further exploration led to the discovery of putative gene clusters in which these seventeen genes were located, and these clusters showed substantial similarity to the clustered portion of carotenoid biosynthetic genes. In addition, fifteen other enzymes associated with C40 backbone synthesis were identified (Appendix A).

A total of 68 PTs (prenyltransferases) and DMATSs (dimethylallyltryptophan synthases) were screened, including DMATS1, which was previously identified in *F. fujikuroi* [54]. There are fifteen homologous sequences to DMATS1 in *Fusarium* species, and their amino acid sequence identity exceeds 64%. It is speculated that these fifteen PTs, similar to DMATS1, are responsible for the trans-prenylation of the N1 position of the indole group (Appendix A). The other two sequences, FOYG_11805 and FACUT_12572, share 77.40% and 76.39% identity, respectively, with 7-DMATS from *A. fumigatus* [55]. It is speculated that these two sequences are also involved in the cis-prenylation of the C7 position of the indole group. Additionally, an uncharacterized PT, FsdK [56], and its two homologues also exhibited a high degree of similarity, with an identity greater than 76%.

### 3.3. Nonribosomal Peptide Biosynthetic Pathway of Pathogenic Fusarium Species

A total of 400 NRPSs were identified in the 35 selected pathogenic *Fusarium* species, including 22 identified NRPSs of *Fusarium* species. The homologues of 22 identified NRPSs were screened based on the cluster analysis of the phylogenetic tree (Figure 7). 

NPS1 is an NRPS from *Histoplasma capsulatum* involved in extracellular siderophore production [57], and four of its paralogues were screened in *Fusarium*, with 54% sequence identity. Further analysis of BGCs found that the gene cluster containing NPS1 was highly similar to six putative *Fusarium*-derived gene clusters (Appendix A). NPS2, the core enzyme of ferricrocin synthesis from *F. graminearum*, has 32 homologous proteins [58]. The comparison found that they have a highly consistent domain composition (Appendix A). NPS6, another NRPS from *F. graminearum* involved in iron-carrier-mediated iron metabolism, also has 32 homologous proteins. The structural features of the remaining 30 homologous sequences are very similar to those of NPS6 [59], except for the partial deletion of the structural domains of FVEG_15442 and FVEG15444 (Appendix A). SidE is an NRPS from *A. fumigatus* involved in siderophore-mediated iron metabolism [60]. In these selected *Fusarium* species, 19 homologous sequences most similar to SidE were screened, and domain feature analysis found that the domain composition of these 19 homologues was highly consistent. In contrast, the N-terminus of SidE lacks a C domain (Appendix A). SidC is another NRPS from *A. fumigatus* involved in iron-carrier-mediated iron metabolism [61], and the six most similar homologous sequences were screened in these *Fusarium* species. Domain analysis showed that the domain composition of NCS54_013740003, FDECE_13287 and FPRO_13979 is highly consistent with SidC, while the remaining three differ significantly from SidC (Appendix A). ESYN1 from *F. oxysporum* was shown to be responsible for the synthesis of cyclic depsipeptide enniatins [62]. Five homologues of ESYN1 were found, and except for FGRM_2752, which lacks an MT domain, the domain composition of the other four homologues was almost identical to ESYN1 (Appendix A). PesF (Afu3g12920) from *A. fumigatus* is a putative ETP toxin synthetase, and five NRPSs with highly similar domain characteristics to AF_NRPS5 were screened (Appendix A). HTS1 is a synthase of HC-toxin from *Cochliobolus Carbonum* [63]. In this selected *Fusarium*, there is a putative NRPS, jgi.p_Fustri1_636519, which is relatively homologous to HTS1. Domain comparison revealed that jgi.p Fustri1_636519 lacks a T domain compared to HTS1 (Appendix A).

The virulence factor beauvericin is synthesized by NRPS encoded by *bbBeas* from *Beauveria bassiana* [64]. Related studies have identified two paralogous homologues of bbBeas in *Fusarium* species, *fpBeas* [65] and *BEA1* [66]. In this work, 17 homologues of BbBeas were identified, and BbBEAS and its 19 paralogous homologues formed distinct clusters in the evolutionary tree (Figure 7). Comparative domain (Figure 8A) and identity analysis (Appendix A) revealed high similarities between BbBEAS and 17 *Fusarium*-derived BEASs, except for partial amino acid deletions in FVEG_16703 and J7337_011263. Beauvericin is the product of a collaboration between BEAS and KIVR in which KIVR converts 2-ketoisovalerate from primary metabolism into one of the initial substrates of BEAS, *D*-2-hydroxyisovalerate (*D*-Hiv). Subsequently, *D*-Hiv and another initial substrate, *L*-phenylalanine (Phe), are catalyzed by BEAS-related structural domains to form a dipeptidol monomer, and three dipeptidol monomers are esterified to form beauvericin [64] (Figure 8B). Comparison of BGCs revealed that *Fusarium*-derived *Bea*_BGCs are highly conserved and similar, with most being more abundant than *bbBea*_BGC which has one more ABC transporter-encoding gene upstream of kivr (Appendix A). 

The NRPS-coding gene *aclP* is the core gene responsible for aspirochlorine biosynthesis identified in *A. oryzae* [67], and its thirteen paralogous genes were screened in these *Fusarium* species. The amino acid sequence identity comparison showed that the identity of AclP with the thirteen paralogous sequences was not more than 40%, and the thirteen paralogous sequences were all higher than 80% with each other (Appendix A). Domain comparison revealed high identities between AclP and its homologues, except for the premature termination of some entries (Appendix A). It is speculated that these NRPSs with intact T-C-A-T-C domains, like AclP, catalyze the formation of *cyclo-*(*L*-Phe-*L*-Phe) (Appendix A). Comparison of BGCs revealed similarities between several genes in *acl*BGC and related genes in the *Fusarium*-derived BGCs (Appendix A). 

*NRPS4* is the gene encoding the cyclic hexapeptide synthase (FGSG_02315) identified in *F. graminearum* [68], and NRPS4 shows higher than 70% identities with its nine paralogues (Figure 9A) that were screened in the present selection of *Fusarium*. Further domain features revealed a high degree of identity between NRPS4 and the nine orthologues, with the exception of FPOAC1_014147 and FPOAC1_013344 (Figure 9B). It is assumed that homologues such as FVRRES_02708, like NRPS4, are also able to synthesize the cyclic hexapeptide fusahexin using related amino acids as substrates (Figure 9C). Comparison of BGCs containing these fusahexin synthases further revealed that they are highly similar (Appendix A). 

The genes encoding the virulence factor linear octapeptide fusaoctaxin A synthase, NRPS5 (FGSG_13878) and NRPS9 (FGSG_10990), were identified in the wheat pathogen *F.* graminearum [69]. Eight *nrps9* paralogous homologues were screened in this work, and all nine NRPSs showed an amino acid sequence identity above 65% (Figure 10A). Structural domain analysis showed a high degree of identity in the structural domains of the remaining eight sequences, with the exception of LC18013491 (Figure 10B). Further BGC comparisons showed that all eight *Fusarium* species, including *Fusarium pseudograminearum*, contained BGCs that were highly similar to the *fg3_54* BGC (Appendix A). Therefore, it is hypothesized that the biosynthetic pathway of fusaoctaxin A is commonly distributed in all nine *Fusarium* species (Figure 10C).

NRPS32 (FPSE_09183) and PKS40 (FPSE_09187) are two core genes identified in *F. pseudograminearum* responsible for the biosynthesis of hybrid compound W493 B [70]. Two paralogous genes were found for NRPS32, and their sequence identity exceeds 88% (Figure 11A). Domain comparison indicated that the three sequences share an identical domain organization (Figure 11B), and further analysis revealed a high degree of similarity in the BGCs where they are located (Figure 11C). Therefore, it can be inferred that these two putative BGCs are also involved in the biosynthesis of W493 B (Figure 11D). Similarly, NRPS7 (FGSG_08209) and PKS6 (FGSG_08208) were identified as two core genes responsible for the biosynthesis of the hybrid compound fusaristatin A in *F. graminearum* [71]. Five paralogous genes were identified for NRPS7, and their sequence identity exceeds 70% (Figure 12A). Domain comparison demonstrated complete domain conservation among these genes (Figure 12B), and further analysis revealed high similarity among the BGCs in which the core genes are located (Appendix A). Therefore, it is speculated that these putative BGCs also have the ability to biosynthesize fusaristatin A (Figure 12C). 

NRPS30 (MAA_10043) is the core gene identified in *Metarhizium robertsii* responsible for the cyclic pentapeptide, sansalvamide [72] (Appendix A). Five paralogous genes of NRPS30 were identified in *Fusarium*, and the sequence identity of NRPS30 with these five NRPSs was not more than 50%, whereas the five NRPSs had more than 70% sequence identity with each other (Appendix A). Domain analysis found that five homologues of NRPS30 lacked the fifth module relative to NRPS30 (Appendix A), and further comparison of BGCs showed that BGCs from *Fusarium* also contained homologous genes of the P450-encoding gene (MAA_10043) (Appendix A). Chry1 (NRPS14, FGSG_11396), the NRPS responsible for the biosynthesis of the alkaloid chrysogines, was identified in *F. graminearum* [73], and the homologues of Chry1 were also screened in seven other *Fusarium* species. Amino acid sequence identity analysis showed that these NRPSs shared more than 60% sequence identity with each other, and domain analysis revealed a high degree of similarity in the composition of the domains of the NRPSs, except for the inappropriately annotated *F. poae* origin NRPS. Further analysis revealed that *chry*BGCs were present in multiple *Fusarium* species and that these BGCs were highly similar (Appendix A).

*GRA1* (*NRPS8*, FGSG_15673) is a core gene in the biosynthesis of the bicyclic toxic lipopeptides gramillins in *F. graminearum* [74]. Two homologous genes of *GRA1*, HYE67_007954 and FGFSG_11659, were found in *F. pose* and another subspecies of *F. graminearum*. Domain comparison showed that the domain composition of GRA1 and HYE67_007954 was highly consistent, while FGFSG_11659 lacked some domains compared to GRA1. BGC comparison showed that *GRA*BGC is highly similar to the BGCs within HYE67_007954 and FGFSG_11659, and it is speculated that these two BGCs may also produce similar cyclic peptide compounds (Appendix A). *Aps1* and *APF1* are core genes responsible for the synthesis of the cyclic tetrapeptides apicidin F and apicidin from *F. semitectum* [45] and *F. fujikuroi* [22], respectively. Their homologues, B0J16DRAFT_375847 and FPOAC1_013755, were screened in two other *Fusarium* species, and the structural compositions of these four NRPSs were highly consistent. Further BGC comparisons showed that *Aps*BGC and *APF*BGC were very similar to the two putative BGCs (Appendix A). 

The cyclic peptide FR901469 was identified as being synthesized by the NRPS FrbI (AN011243_029940) encoded by the unknown fungal species No. 11243 [75], and several frbI-like genes were screened in *Fusarium* strains. A comparison of the structural domains revealed that the NRPSs from the *Fusarium* species lacked most of the domains compared to FrbI (Appendix A). NRPSs from three different *Fusarium* strains, J7337_003370, FNAPI_7159 and FVRRES_13918, showed a certain degree of conservation with the PKS-NRPS1 (FFUJ_02219) [57] from *F. fujikoroi*. Comparison of the structural domains revealed that the amino acid sequences of these three NRPSs were similar to the NRPS modules of the hybrid enzyme PKS-NRPS1 (Appendix A). The NRPS FMAN_12219 from *F. mangiferae* showed some similarity to another hybrid enzyme FUS1 (FFUJ_10058) derived from *F. fujikuroi* [23], and a comparison of the structural domains also revealed a high degree of correspondence between the amino acid sequence of FMAN_12219 and the NRPS module of the hybrid enzyme PKS-NRPS1 (Appendix A).

### 3.4. Polyketide Biosynthetic Pathway of Pathogenic Fusarium Species

In this comprehensive analysis of pathogenic *Fusarium*, a total of 522 PKSs were identified in 35 carefully selected strains. These PKSs are vital enzymes involved in the synthesis of polyketide compounds, many of which play a crucial role in the virulence and pathogenicity of these fungal species. Through the utilization of phylogenetic tree clustering analysis, twenty-three distinct PKS clades were functionally identified (Appendix A). 

Gibepyrones are fungal toxins that have been isolated from the rice pathogen *F. fujikuroi*, and their biosynthetic pathway has also been elucidated in *F. fujikuroi* [76]. The PKS-encoding gene GPY1 is considered to be the core gene involved in the biosynthesis of gibepyrones, especially gibepyrone A. Notably, the cluster analysis revealed the widespread presence of GPY1 homologues among the 35 selected pathogenic *Fusarium* species. Amino acid sequence analysis revealed that GPY1 shares more than 75% sequence identity with its homologues (Figure 13A), and further investigation of the domain composition of these PKSs indicated high conservation (Figure 13B). The putative BGCs containing GPY1 homologues are commonly found in *Fusarium* species, with striking similarities to the *GPY*BGC (Appendix A). Fusarubins, another class of polyketides, have also been isolated from *F. fujikuroi*, and their biosynthetic pathway has been identified in this fungal strain as well [77]. The BGC responsible for fusarubins consists of six genes, with *fsr1* identified as the core gene encoding for the creation of the fusarubin skeleton, specifically 6-*O*-methylfusarubin [77]. The homologues of *fsr1* were found to exist in all 35 selected pathogenic *Fusarium* species, and their amino acid sequences exhibited a remarkable level of identity exceeding 75% (Figure 14A). Additionally, the analysis of domain compositions indicated high similarity (Figure 14B). The *fsr*-like BGCs were found to be widely present across pathogenic *Fusarium* species (Appendix A). 

Fusaric acid, a notorious mycotoxin known to cause extensive damage to plants, has also been the focus of BGCs identified in various *Fusarium* species [11], including *F. fujikuroi*. The *FUB1* gene, which encodes a highly reductive PKS, is considered a key gene within the *FUB*BGC [78]. Homologues of FUB1 were found to have a wide distribution in over twenty *Fusarium* species, displaying up to 93% sequence identity at the amino acid level (Figure 15A) with highly conserved domain structures (Figure 15B). Further analysis revealed the presence of the predicted *FUB*BGC in several pathogenic *Fusarium* species, with significant similarity to previously identified *FUB*BGCs (Appendix A).

Bikaverin, a strikingly pigmented compound, was initially identified in cultures of *F. lycopersici* and *F. vasinfectum* [79]. The BGC responsible for bikaverins was discovered in *F. fujikuroi* [80]. The initiation of *bikaverin* biosynthesis is mediated by the PKS-encoding gene *bik1*, and Bik1 utilizes acetyl-coenzyme A and malonyl-coenzyme A to produce the bicyclic precursor of bikaverin, referred to as pre-bikaverin. Nineteen homologues of Bik1 were identified, with amino acid sequence identities exceeding 81% (Figure 16A). Analysis of the domain composition revealed striking similarities between Bik1 and its homologues (Figure 16B). Further investigation uncovered the widespread presence of predicted bikBGs in pathogenic *Fusarium* species, which exhibited high similarity to the identified *bik*BGC (Appendix A). *FmFPY1* (*FmPKS40*), a key gene involved in the biosynthesis of fusapyrone and deoxyfusapyrone [70], has been found to have fifteen homologues. The sequence identities between FmFPY1 and its homologues surpass 81%. Despite the absence of the C-terminal domain in J7337_000001 and FGLOB1_11207, the other PKSs share remarkably similar domain compositions. Putative BGCs containing *FmFPY1* homologues have been identified in multiple pathogenic *Fusarium* species, which show a high degree of similarity (Appendix A).

The gene *fogA*, derived from *A. ruber*, serves as the core gene responsible for flavoglaucin biosynthesis [81]. Several homologues of FogA have been discovered in pathogenic *Fusarium* species. The sequence identity between FogA and its homologues exceeds 50%, while the *Fusarium*-derived homologues show more than 90% identities (Figure 17A). A comparison between *fog*BGC and putative BGCs containing *fogA* homologues from *Fusarium* indicates some similarity, whereas the *Fusarium*-derived putative BGCs show high similarity (Figure 17B). Furthermore, putative *fog*BGCs were discovered in twelve pathogenic *Fusarium* species, which exhibit high similarity to the *fog*BGC (Appendix A). SdnO, a PKS identified in the BGC responsible for sordarin in *Sordaria araneosa*, plays a crucial role in the synthesis of the glycolipid sidechain of the sordarin structure [82]. Screening of pathogenic *Fusarium* species led to the discovery of five homologues to SdnO (Appendix A). Although the identity between SdnO and these homologues does not exceed 40%, the identities among these homologues themselves surpass 60%. Further comparisons revealed a remarkable similarity in domain features between four of these homologues and SdnO (Appendix A).

YWA1 serves as an intermediary compound in the biosynthesis of aurofusarin, a pigment toxin found in *F. graminearum* [2]. The biosynthesis of aurofusarin is initiated by PKS12 [83], which is encoded by the *fus*BGC. Through sequence analysis, eight PKS12 homologues with a sequence identity exceeding 75% were identified (Figure 18A). Additionally, these homologues exhibited significant domain similarity (Figure 18B). Examination of predicted BGCs containing PKS12 revealed their presence in pathogenic *Fusarium* species, further highlighting their similarity to the *fus*BGC (Figure 18C). Hence, it can be inferred that aurofusarin is a commonly produced pigment toxin in these fungi. Depudecin, a linear polyketide with eleven carbon atoms, was isolated from the pathogenic fungus *Alternaria brassicicola* [84]. The core gene responsible for the biosynthesis of depudecin is *DEP5* [84]. Screening identified nine homologous sequences of DEP5 that share more than 65% sequence identity and have similar domain compositions. This suggests a conserved function across these homologues. Furthermore, putative *DEP*BGCs were discovered in eight pathogenic *Fusarium* species, and they are highly similar to the *DEP*BGC (Appendix A).

*PKS6* is considered one of the pivotal genes involved in the biosynthesis of the cyclic peptide fusaristatin A [71]. This gene operates in conjunction with another core gene, NRPS7, to synthesize fusaristatin A (Figure 12). Five highly similar homologues of PKS6 have been identified based on both amino acid sequence identity and domain composition (Appendix A). Similarly, PKS40 and NRPS32 form another pair of synergistic core genes responsible for the production of W493 B [71] (Figure 11), and two homologues of PKS40 with identical domain structures were identified (Appendix A). Furthermore, in *F. fujikuroi*, *PKS19* serves as the core gene for the biosynthesis of α-pyrones (fujikurins). Three homologues of PKS19 have been identified in pathogenic *Fusarium* species, and their domain features closely resemble each other (Figure 19A). Additionally, the comparison of BGCs revealed a remarkable similarity between the presumed BGCs containing PKS19 homologues and the fujikurin BGC (Figure 19B).

*Alt5* has been recognized as the core gene responsible for the biosynthesis of alternapyrone in *A. solani* [85]. Four PKSs were identified that shared more than 70% identity with Alt5. These PKSs shared consistent domain features, and their corresponding BGCs displayed a notably high similarity (Appendix A). In addition, the core gene *sol1*, which is responsible for the synthesis of solanapyrone, was identified in *A. solani* [86]. Three homologues of sol1 were screened in *Fusarium* species, and these four PKSs exhibited very similar domain characteristics (Appendix A). *DpfgA*, identified in *F. graminearum*, functions as a core gene responsible for the polyketone part of subglutinol biosynthesis [43]. Through screening, four homologues of DpfgA with highly similar domain features were identified in other pathogenic *Fusarium* species (Appendix A). *FSL1*, the core gene responsible for fusarielin biosynthesis, was also identified in *F. graminearum* [87]. It cooperates with FSL5 to complete the backbone synthesis of fusarielins. Five homologues of FSL1 were screened, and they displayed a high degree of similarity in their domain compositions (Appendix A). Moreover, predicted *FSL*BGCs were discovered in the corresponding strains, exhibiting considerable similarity to the *FSL*BGC (Appendix A). Another pair of genes, *bet1* and *bet3*, function collaboratively to form a polyketone skeleton [88], with Bet1 belonging to the type I HR PKS. A *bet*-like BGC was identified in *F. decemcellulare*, wherein three genes displayed significant homology with *bet1*, *bet3* and *bet4*, respectively (Appendix A). The core gene *G433*, responsible for the synthesis of 1233A, was identified in *Fusarium* sp. RK97-94 [89,90]. Through screening, four homologues of G433 were identified in the 35 selected pathogenic *Fusarium* species. G433 and these homologues show high similarity in domain composition, and the corresponding BGCs display a significant homology (Appendix A). 

The gene *FUM1*, which encodes the PKS involved in fumonisin synthesis, is considered to be the key gene in this process [91,92]. Six homologues of FUM1 have been identified by screening. FUM1 and its homologues share not only a high degree of similarity in their amino acid sequences (Appendix A) but also a close resemblance in the composition of their domains (Appendix A). Putative *FUM*BGCs were identified in related species and showed striking similarity to *FUM*BGCs (Appendix A). In the pathogenic *G. zeae*, two core genes, *zea1* (pks13) and *zea2* (pks4), were identified as essential for zearalenone synthesis [93,94]. These two genes work together to produce the linear backbone structure of zearalenone. Several putative *zea*BGCs have been identified in pathogenic *Fusarium* species, and their core enzymes, which are homologues of Zea1 and Zea2, showed remarkable domain similarities to Zea1 and Zea2 (Appendix A). In addition, a pair of synergistic PKS-encoding genes, pkhA and pkhB, were identified as the core genes responsible for alternariol biosynthesis [95]. Several putative *phk*BGCs were screened in the pathogenic *Fusarium* species, and their core enzymes, which are homologues of PkhA and PkhB, showed high structural similarity to PkhA and PkhB (Appendix A). 

### 3.5. PKS-NRPS Biosynthetic Pathway of Pathogenic Fusarium Species

In the field of mycology, the colocalization of PKS and NRPS genes in fungi leads to the formation of PKS-NPS hybrid enzymes. These enzymes consist of both PKS units, which contain various domains such as KS, AT, DH, ME, KR and ACP, as well as NRPS units, which consist of A, T and C domains. Within this complex enzyme system, the PKS units primarily mediate reactions involved in elongating carbon chains, while the NRPS units utilize the A domain to selectively activate specific amino acids and load the resulting aminoacyl residues onto the T domain. Once the entire polyketide chain assembly is complete, the C domain facilitates the fusion of the polyketide chain with the activated amino acid residues, ultimately resulting in the production of amide-derived compounds. The first characterized PKS-NRPS was discovered in the genus *Fusarium*, and to date, a total of six PKS-NRPSs have been deciphered from different *Fusarium* species. Evolutionary analysis of 88 PKS-NRPSs screened from the 35 pathogenic *Fusarium* species and five identified PKS-NRPSs-derived from non-*Fusarium* species allowed them to form distinct clusters (Figure 20).

One notable example of a PKS-NRPS hybrid phytotoxin is Fusarin C, which was identified in maize infected with the plant pathogenic fungus *F. moniliforme* back in 1981 [96]. The core genes responsible for the biosynthesis of fusarin C, FusA or Fus1, were subsequently identified in *F. moniliforme* [97] and *F. fujikuroi* [23], respectively. Interestingly, eighteen homologues of FusA and Fus1 have been found in other pathogenic *Fusarium* species, and these twenty sequences make up the largest cluster of the PKS-NRPS collection. These hybrid enzymes share more than 70% of the amino acid sequence with each other (Figure 21A). Apart from five sequences that contain additional ER domains, the domain composition of the remaining sequences is consistent with that of FusA and Fus1 (Figure 21B). It is hypothesized that these homologous sequences, like FusA and Fus1, synthesize pre-Fusarin C with high homoserine, malonyl-CoA and S-adenosyl-*L*-methionine (SAM) as substrates (Figure 21C). Furthermore, putative *Fus*BGCs have been discovered in eighteen additional pathogenic *Fusarium* species, which show significant similarity to the *Fus*BGCs associated with the biosynthesis of Fusarin C (Appendix A). Another related compound, lucilactaene, which is a structural analogue of Fusarin C, has been isolated from *Fusarium* sp. RK97-94. The core gene responsible for lucilactaene biosynthesis, *luc5* [90], has been found in four homologues in the pathogenic *Fusarium* species, and these five sequences share over 92% identity (Figure 22A). Not only do their domain features bear a significant similarity to Luc5 (Figure 22B), but the BGCs containing PKS-NRPSs in these species also show high similarity to the *luc*BGC (Appendix A). Presumably, these homologous sequences, like Luc5, synthesize analogues of pre-Fusarin C with the same substrates as FusA and Fus1 (Figure 22C).

Through the application of cluster analysis, we have discovered eighteen novel PKS-NRRSs that form the second-largest clade within the hybrid collection. These newly identified sequences display divergence from previously characterized PKS-NRRSs, as evidenced by them sharing a less than 40% sequence similarity and identity. However, there is a striking intrasequential congruence, with identity values reaching up to 80% (Appendix A). Structural alignment reveals an almost perfect homology in terms of domain composition (KS-AT-DH-MT-KR-ACP-C-A-T-R-ER) among these newly identified sequences (Appendix A). Similarly, the putative BGCs in which these PKS-NRRSs serve as core genes also exhibit a significant similarity (Appendix A). Based on these findings, we propose that these newly discovered PKS-NRRSs may represent a previously unknown class of hybrid enzymes. Additionally, it is conceivable that the BGCs containing these hybrid enzymes may play a vital role in the synthesis of novel fungal toxins.

Sambutoxin, a mycotoxin, was initially discovered in the potato pathogen *F. sambucinum* [98]. The core gene responsible for the biosynthesis of sambutoxin, known as *smbB*, was identified in *F. commune* [25]. Fourteen sequences similar to SmbA have been found in other pathogenic *Fusarium* species, with over 75% similarity in their amino acid sequence (Figure 23A). Further analysis of the structure showed a high consistency in the domain features among these sequences (Figure 23B), suggesting that these SmbA homologues use phenylalanine, acetyl-CoA, malonyl-CoA and SAM as substrates to synthesize a hybrid scaffold, which serves as the precursor for mycotoxins (Figure 23C). The putative samBGCs were identified in the corresponding fifteen pathogenic *Fusarium* species, and these putative BGCs share a significant resemblance to *smb*BGCs (Appendix A). 

Equisetin and trichosetin, naturally occurring tetramic acids derived from PKS-NRPS, are phytotoxic and exhibit cytotoxic effects. These compounds are produced by the pathogenic *Fusarium*. The core genes involved in their biosynthesis, *fsa1* [99], *eqiS* [24] and *FFUJ_02219* [100], have been identified in *Fusarium* sp. FN080326, *F. heterosporum* and *F. fujikuroi*, respectively. A total of fifteen sequences similar to equisetin synthetase were identified in pathogenic *Fusarium* species, and these eighteen sequences share more than 75% sequence identity among themselves (Figure 24A). Domain analysis indicates that these eighteen sequences have highly similar structural features (Figure 24B). Based on the known equisetin synthetases, it is suggested that these hybrid enzymes also employ serine and coenzyme A derivatives in the synthesis of equisetin compounds (Figure 24C). The putative BGCs for equisetins were identified in the fifteen respective pathogenic *Fusarium* species, and these BGCs showed a high level of similarity to each other (Appendix A).

The core gene responsible for the biosynthesis of ilicicolin H in *Penicillium variabile*, *iccA*, is a hybrid gene consisting of multiple modules [101]. IccA and IccB work together to create the hybrid scaffold [101]. Six homologues of IccA have been identified in pathogenic *Fusarium* species. The amino acid sequence identity between IccA and these six *Fusarium*-derived PKS-NRPSs ranges from 50% to 60%, while the identity among the six *Fusarium*-derived PKS-NRPSs themselves exceeds 70% (Figure 25A). The alignment of their domains shows a high congruence between the domain composition of IccA and those of the six *Fusarium*-derived PKS-NRPSs (Figure 25B). Therefore, it is hypothesized that these six *Fusarium*-derived PKS-NRPSs, similar to IccA, utilize tyrosine, SAM and coenzyme to synthesize tetramic acid intermediates (Figure 25C). Based on the predictions from antiSMASH, putative *icc*BGCs have been identified in the corresponding six pathogenic *Fusarium* species, and these BGCs show remarkable similarity (Appendix A).

The PKS-NRRS FsdS, derived from *F. heterosporum* [56], consists of ten domains (KS-AT-DH-MT-KR-ACP-C-A-T-R), where the A domain is responsible for the activation of *L*-tyrosine [56]. A comparison of the amino acid sequences of FDECE_13779 and FGRMN_3691, two PKS-NRSs from pathogenic *Fusarium* species, showed that they share a significant 72% sequence identity with FsdS (Appendix A). The domain alignment of these three sequences indicated that the structural features were indeed identical (Appendix A). Further comparisons carried out on BGCs revealed a significant similarity between the BGCs containing FDECE_13779 and FGRMN_3691 and *fsd*BGC (Appendix A). In addition, ACE1 is a key gene involved in the biosynthesis of an avirulence signaling compound in the rice pathogen *Magnaporthe oryzae* [102]. Notably, ACE1 also shows similarity to two other PKS NRRSs from pathogenic *Fusarium* species, namely, FDECE_13779 and FGRMN_3691. *ACE*BGC showed some similarity to the BGCs containing FDECE_13779 and FGRMN_3691, with both the core gene (Appendix A) and related functional genes showing high homology (Appendix A). Furthermore, the *thnA* gene identified in *Trichoderma harzianum* serves as a core gene for the synthesis of trihazones [103]. Interestingly, there is a 69% amino acid sequence identity between ThnA and FSARRC_13765 from *F. Sarcochroum*. Not only do FSARRC_13765 and ThnA share highly similar domain compositions (Appendix A), but they also show considerable similarity within the BGCs in which they are located (Appendix A). The PKS-NRPS encoding gene, *chgG*, was identified in *Chaetomium globosum* [104]. CghG shares more than 56% amino acid sequence identity with FMUND_12554, and there is some similarity in their domain compositions (Appendix A). Finally, two novel PKS-NRSs have been identified in *Fusarium*, namely, LCI18_013989 and jgi.p_Fustri_620762. The structural composition of LCI18_013989 consists of the domains KS-AT-DH-MT-KR-ACP-C-A-T-R, whereas jgi.p_Fustri_620762 contains the domains KS-AT-DH-MT-KR-ACP-C (Appendix A). To better understand their functions, these novel PKS-NRSs require further investigation by heterologous expression.

## 4. Discussion

*Fusarium* is a widely distributed filamentous fungus worldwide, and taxonomic studies have identified approximately 400 phylogenetically distinct species in the genus *Fusarium* (https://www.Fusarium.org, accessed on 5 July 2023). Although the genus *Fusarium* is not the most abundant filamentous fungus, *Fusarium* is one of the filamentous fungal groups most closely associated with agricultural production and human health. Mycotoxins produced by pathogenic *Fusarium* species, such as DON, fumonisin B1, T-2 toxin, zearalenone and fumonisins, cause scab, foot rot and head blight on crops and food poisoning in humans and animals F [1]. Advances in sequencing technology have enabled more *Fusarium* genomes to be sequenced, and the deepening of biosynthesis research has continuously revealed the biosynthesis pathways of *Fusarium* mycotoxins. These results provide more convenience for the understanding and cognition of mycotoxins.

The statistical analysis of BGC types predicted by antiSMASH found that the number of various BGCs in the 35 pathogenic *Fusarium* species showed a convergence, that is, the number of NRPSs was the largest, while the number of hybrid enzymes was the least. Among these predicted NRPSs, nearly half (362) of the sequences are actually NRPS-like (Appendix A). Among the real 400 NRPSs, a small number of NRPSs are siderophore-associated transport peptides, such as NPS2 and NPS6 distributed in 32 pathogenic *Fusarium* species (Figure 7). Most NRPSs are the core enzymes of toxic peptide biosynthesis, such as beauvericin synthase distributed in nineteen species. As far as PKSs are concerned, the BGCs for gibepyrones (Figure 13), fusarubins (Figure 14) and bikaverins (Figure 16), three typical polyketide compounds, are almost widely present in these 35 pathogenic *Fusarium* species. In addition, homologous sequences of several PKSs identified in non-*Fusarium* species, such as FogA (Figure 17) and SdnO (Appendix A), were identified in multiple pathogenic *Fusarium* species, and the corresponding BGCs were highly similar. This finding suggests interspecies conservation in the production of these toxins. The production of toxins by microorganisms is usually considered to be a defense against external enemies and an adaptation to the environment, whereas the prevalence of such toxin BGCs may be the result of horizontal gene transfer [105,106,107,108].

In the cluster analysis of PKS-NRPSs, the clade within FusA and Fus1 is the largest cluster among the hybrid enzymes, and the clade within LUC5 is the closest to it (Figure 20). The structural similarity between Fusarin C catalyzed by FusA or Fus1 (Figure 2) and LUC5 catalytic product (Figure 22) reflects their sequence similarity. In addition, a new PKS-NPRS group with eighteen members was found (Figure 19). Although FsdS is the closest relative to this group on the phylogenetic tree, its domain composition is significantly different (Appendix A). PKS-NRPS hybrid enzymes are the main contributors to hybrid toxins, but they are not the only ones. The cooperation of PKS and NRPS also produces hybrid toxins, such as PKS40 and NRPS32 to produce W493 B (Figure 11) and PKS6 and NRPS7 to produce fusaristatin A (Figure 12). However, the hybrid product created by this synergistic effect does not possess the nitrogen-containing five-membered heterocyclic pyrrolidone contained in the natural hybrid compound, such as Fusarin C (Figure 21).

The hazards of mycotoxins to human production activities are self-evident, and there is a great deal of concern about how to effectively prevent these toxicities. Inhibiting the production of these toxins in the causative organisms is a highly effective preventive program that addresses the toxin problem at its source. The results of the present study provide support for such a program. For example, BeaS, the core enzyme for the biosynthesis of beauvericins, which is present in eighteen pathogenic *Fusarium* species and a variety of other filamentous fungi (Figure 8), could be a target for the development of antimicrobial drugs for the inhibition of beauvericin production.

## 5. Conclusions

Although the increasing number of reported *Fusarium* genomes and growing number of biosynthetic studies have led to a better understanding of *Fusarium* mycotoxin production over time, the species distribution and species specificity of *Fusarium* mycotoxin production are often overlooked. Here, we used bioinformatic methods to systematically investigate the core genes involved in the secondary metabolite biosynthesis of 35 pathogenic *Fusarium* species and identified different types of terpene synthesis (cyclization) enzymes and the distribution of more than twenty kinds of NRPSs and PKSs, and more than ten kinds of PKS-NRPSs in these species. This study found that the biosynthesis core genes and corresponding BGCs of gibepyrones, fusarubins, bikaverins and other mycotoxins are almost universally present in these pathogenic *Fusarium*. This study demonstrates the diverse potential of pathogenic *Fusarium* to biosynthesize toxins. These findings provide new insights into the toxins produced by pathogenic *Fusarium* and further provide a theoretical basis for the use of biotechnology to control the production of related toxins.

## Figures and Tables

**Figure 1 jof-09-00850-f001:**
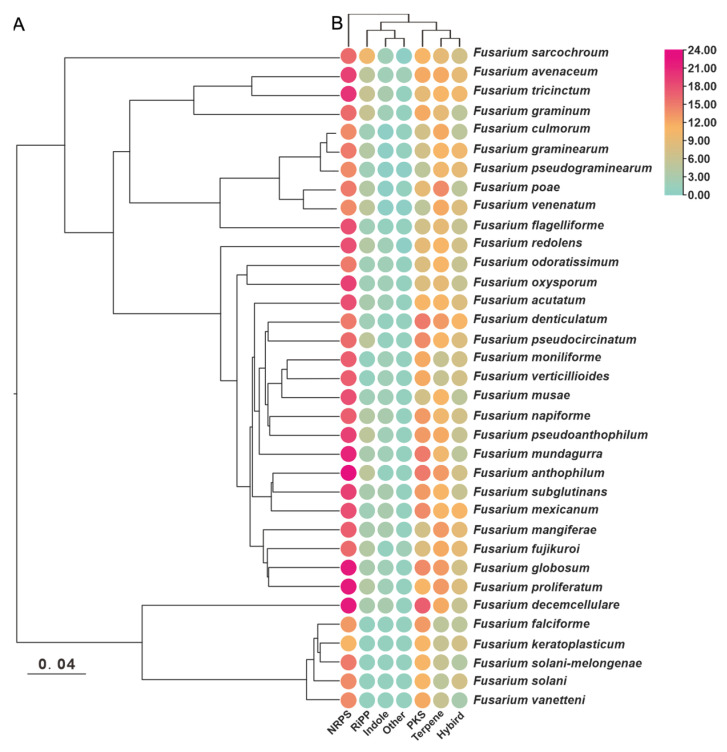
Distribution of six different types of gene clusters in 35 pathogenic *Fusarium* species. (**A**) Evolutionary relationships of the 35 pathogenic *Fusarium* species were constructed based on orthologous single-copy genes. (**B**) The BGC types were counted to visualize based on antiSMASH predictions.

**Figure 2 jof-09-00850-f002:**
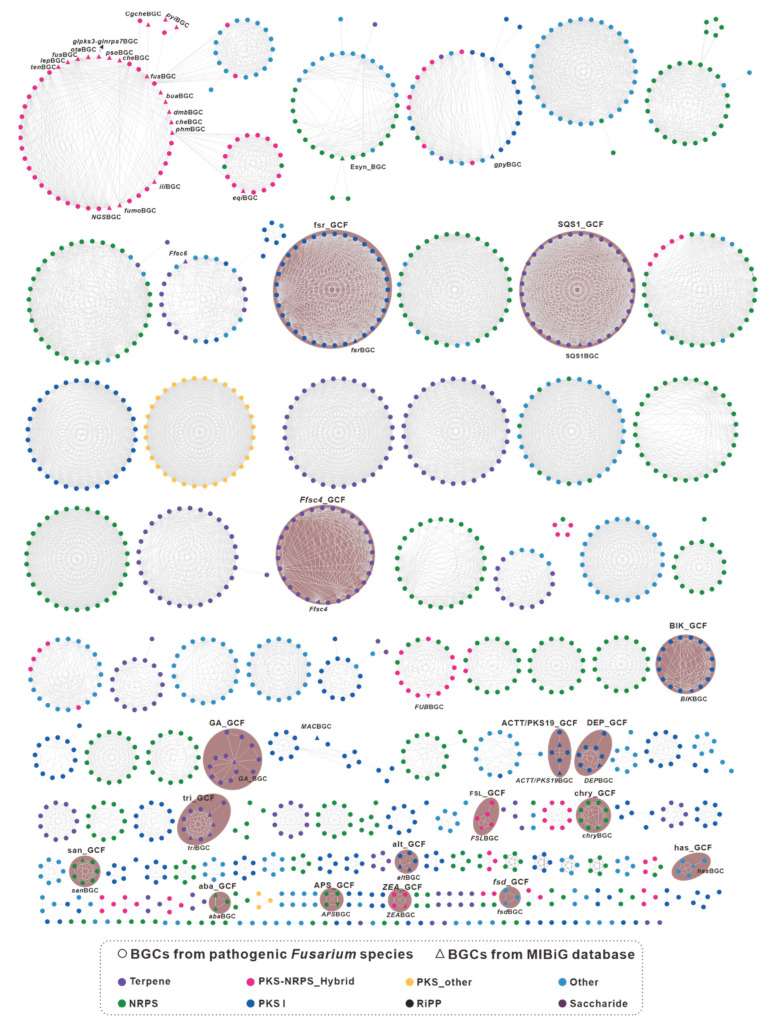
GCF network of the 1733 predicted biosynthetic gene clusters (BGCs) from 35 pathogenic *Fusarium* species calculated by the BiG-SCAPE pipeline and visualized with Cytoscape. The hollow triangles represent the BGCs from the MIBIG database, whose names are labeled near the corresponding hollow triangles. The network with a shaded background represents the identifiable GCFs.

**Figure 3 jof-09-00850-f003:**
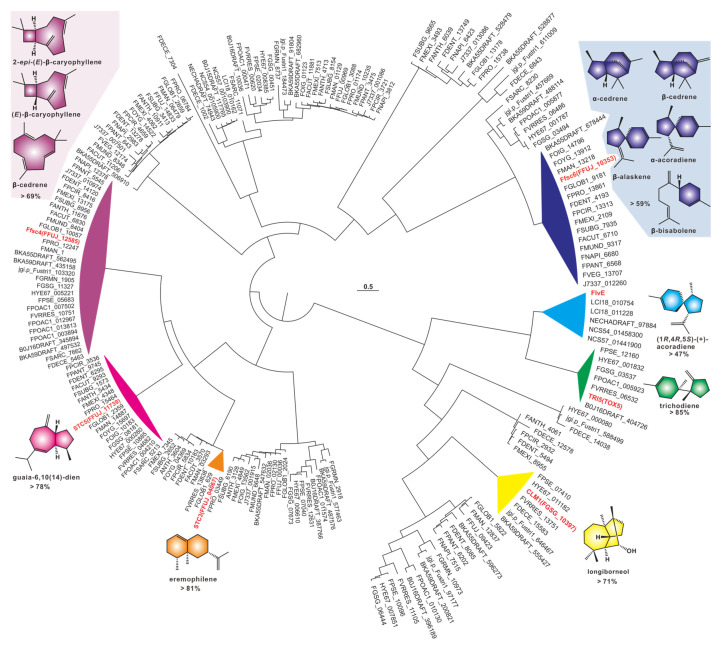
Cluster analysis of 209 sesquiterpene sequences. The identified sequences are marked in red, and groups containing identified sequences are highlighted with a colored background. The fill color of products of identified sequences corresponds to the background color of the group in which they are located.

**Figure 4 jof-09-00850-f004:**
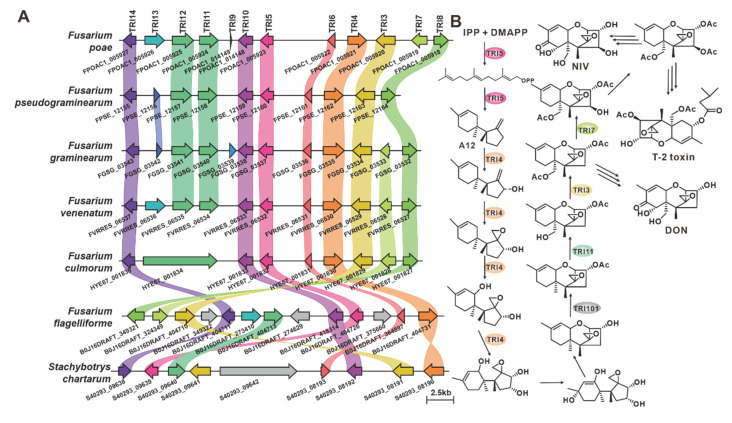
Biosynthesis of trichodiene-type analogues in pathogenic *Fusarium* species. (**A**) Comparison of *tri*BGCs from different species: homologous genes are connected by a band of the same color. (**B**) Biosynthetic pathways of trichodiene-like toxins: Tri5 indicates trichodiene synthase; Tri4 and Tri11 indicate cytochrome P450 monooxygenases; Tri6 indicates regulatory protein; Tri3, Tri7 and Tri101 indicate acetyltransferases.

**Figure 5 jof-09-00850-f005:**
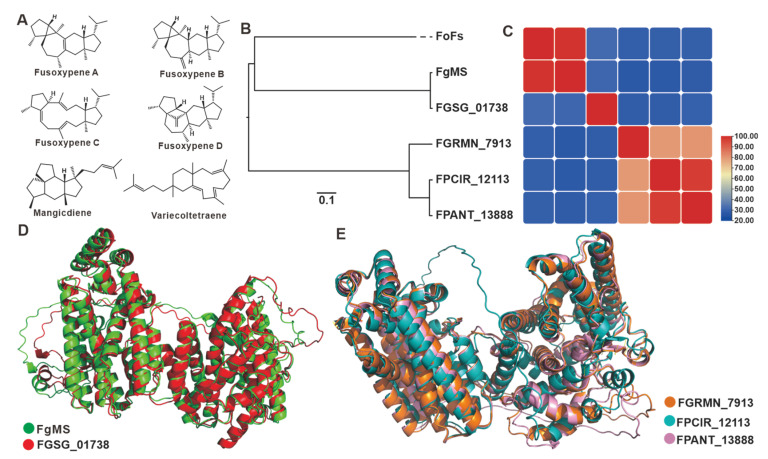
*Fusarium*-derived sesterterpenoids and sesterterpene synthases. (**A**) Structures of representative sesterterpenoids from *Fusarium* species. (**B**) Clustering analysis of chimeric sesterterpene cyclases. (**C**) Amino acid sequence identity analysis of chimeric sesterterpene cyclases. (**D**) Structural comparison of the predicted three-dimensional structures of FgMS and FGSG_01378. (**E**) Structural comparison of the predicted three-dimensional structures of FGRMN_7913, FPCIR_12113 and FPANT_13888.

**Figure 6 jof-09-00850-f006:**
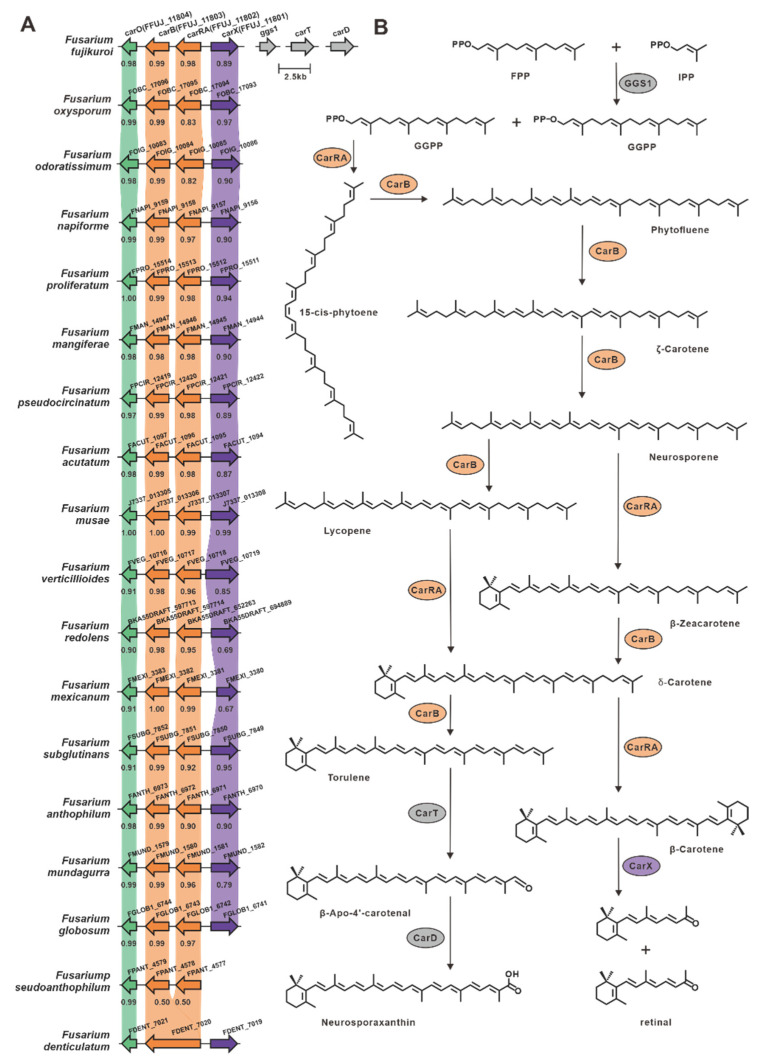
Biosynthesis of carotenoids in *Fusarium* species. (**A**) Comparison of *car*BGCs from different *Fusarium* species: homologous genes are connected by a band of the same color, and the identity value between the two homologous proteins is shown below the previous gene. (**B**) Biosynthetic pathways of carotenoids: CarRA indicates bifunctional phytoene synthase/carotene cyclase; CarB indicates phytotene desaturase; CarT indicates cleaving oxygenase; CarD indicates aldehyde dehydrogenase; and CarX indicates oxygenase.

**Figure 7 jof-09-00850-f007:**
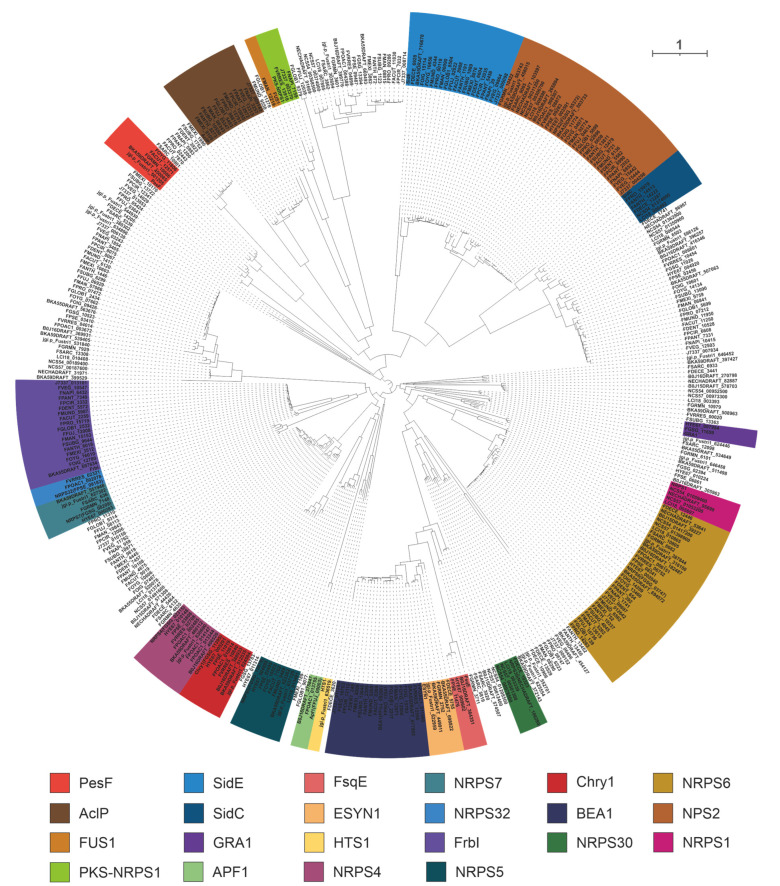
Clustering analysis of NRPSs based on evolutionary trees. Different colored shaded backgrounds represent different clusters, and these clusters with shaded backgrounds contain identified NRPSs.

**Figure 8 jof-09-00850-f008:**
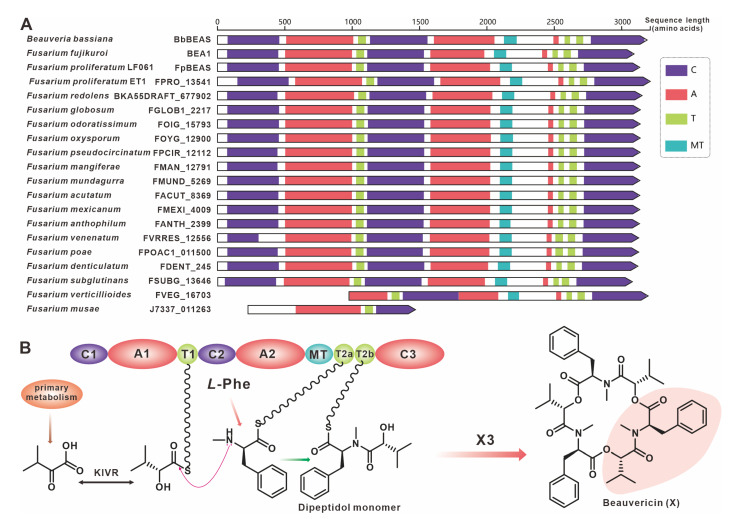
Beauvericin biosynthesis. (**A**) Domain comparison of BbBEAs for the biosynthesis of beauvericin: different colored regions represent different domains. (**B**) Model of the biosynthesis of beauvericin.

**Figure 9 jof-09-00850-f009:**
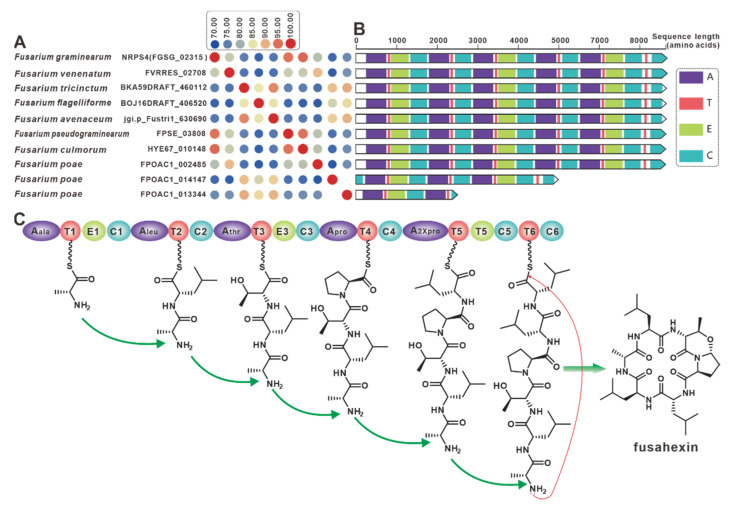
Fusahexin biosynthesis. (**A**) Comparison of the amino acid sequence identity of NRPS4 and its homologues. (**B**) Domain comparison of core genes for the biosynthesis of fusahexin: and different colored regions represent different domains. (**C**) Model of the biosynthesis of fusahexin.

**Figure 10 jof-09-00850-f010:**
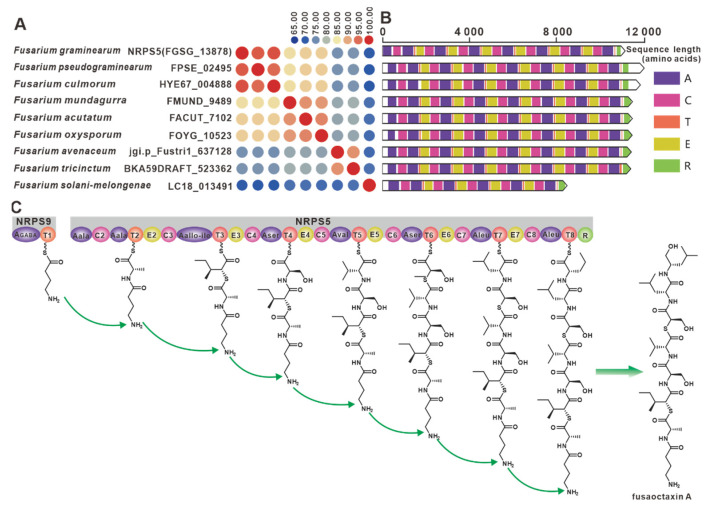
Fusaoctaxin A biosynthesis. (**A**) Comparison of the amino acid sequence identity of *nrps9* and its homologues. (**B**) Domain comparison of NRPS9 for the biosynthesis of fusaoctaxin A: different colored regions represent different domains. (**C**) Model of the biosynthesis of fusaoctaxin A.

**Figure 11 jof-09-00850-f011:**
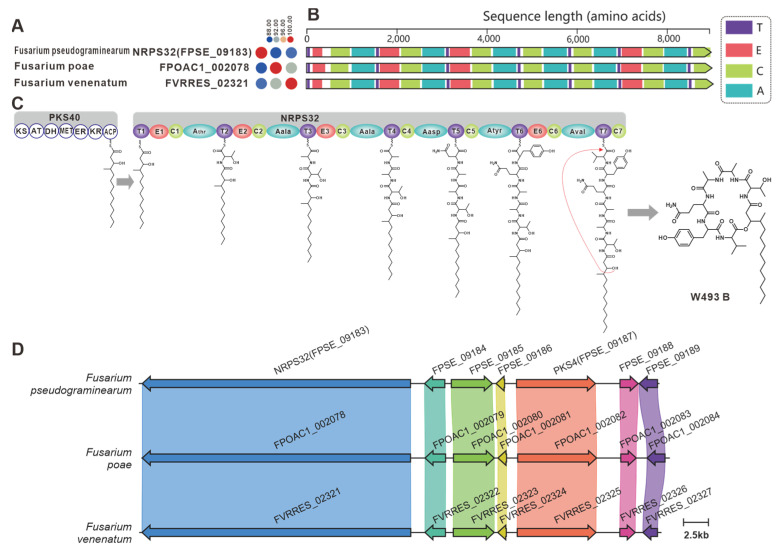
W493 B biosynthesis. (**A**) Comparison of the amino acid sequence identity of NRPS32 and its homologues. (**B**) Domain comparison of NRPS32 for the biosynthesis of W493 B: different colored regions represent different domains. (**C**) PKS40 and NRPS32 collaborative model of the biosynthesis of W493 B. (**D**) Comparison of the BGC for W493 B and two putative BGCs: homologous genes are connected by a band of the same color.

**Figure 12 jof-09-00850-f012:**
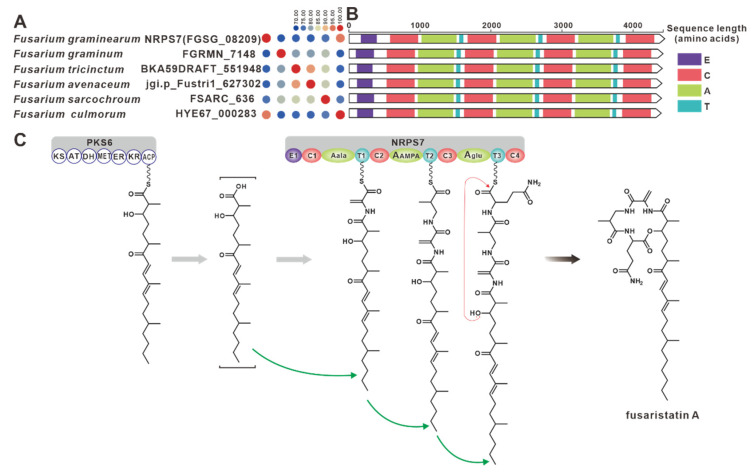
Fusaristatin A biosynthesis. (**A**) Comparison of the amino acid sequence identity of NRPS7 and its homologues. (**B**) Domain comparison of NRPS7 for the biosynthesis of fusaristatin A: different colored regions represent different structural domains. (**C**) PKS6 and NRPS7 collaborative model of the biosynthesis of fusaristatin A.

**Figure 13 jof-09-00850-f013:**
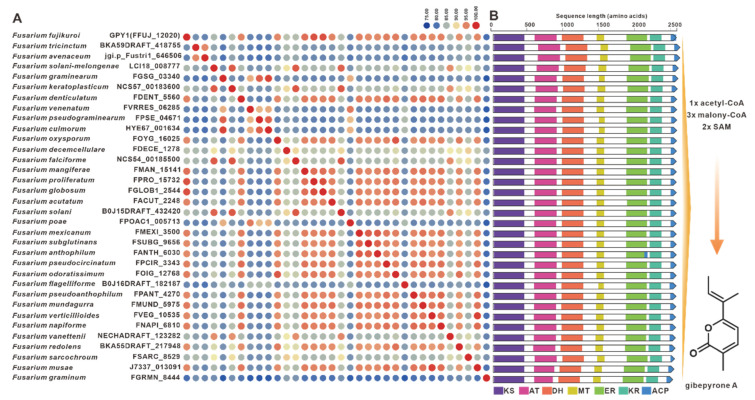
Gibepyrone A biosynthesis. (**A**) Comparison of the amino acid sequence identity of GPY1 and its homologues. (**B**) Domain comparison of GPY1 for the biosynthesis of gibepyrone A: different colored regions represent different domains.

**Figure 14 jof-09-00850-f014:**
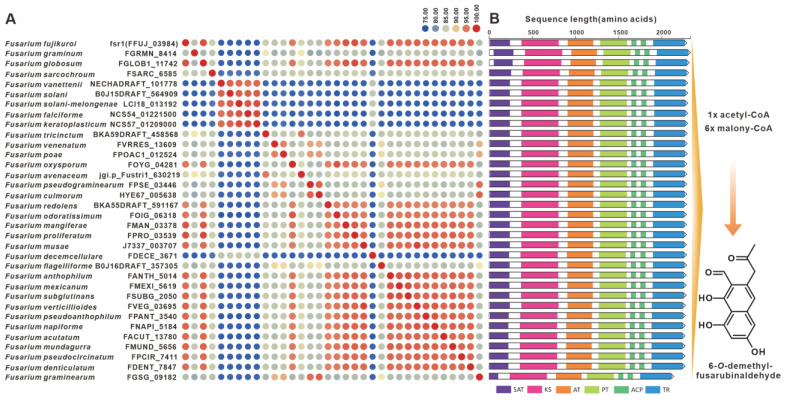
6-*O*-methylfusarubin biosynthesis. (**A**) Comparison of the amino acid sequence identity of Fsr1 and its homologues. (**B**) Domain comparison of Fsr1 for the biosynthesis of 6-*O*-methylfusarubinfusarubin: different colored regions represent different domains.

**Figure 15 jof-09-00850-f015:**
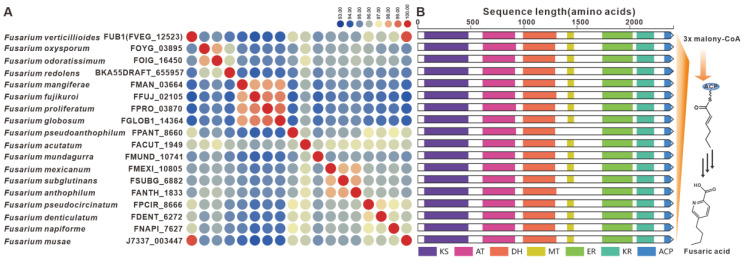
Fusaric acid biosynthesis. (**A**) Comparison of the amino acid sequence identity of FUB1 and its homologues. (**B**) Domain comparison of FUB1 for the biosynthesis of fusaric acid: different colored regions represent different domains.

**Figure 16 jof-09-00850-f016:**
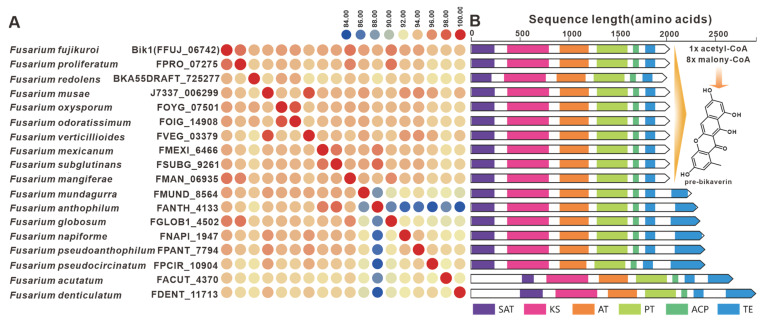
Pre-bikaverin biosynthesis. (**A**) Comparison of the amino acid sequence identity of Bik1 and its homologues. (**B**) Domain comparison of Bik1 for the biosynthesis of pre-bikaverin: different colored regions represent different domains.

**Figure 17 jof-09-00850-f017:**
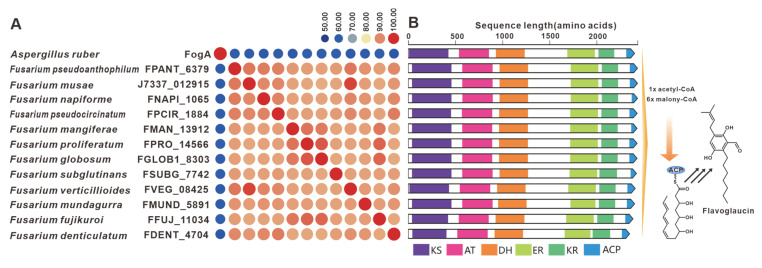
Flavoglaucin biosynthesis. (**A**) Comparison of the amino acid sequence identity of FogA and its homologues. (**B**) Domain comparison of FogA and its homologues. different colored regions represent different domains.

**Figure 18 jof-09-00850-f018:**
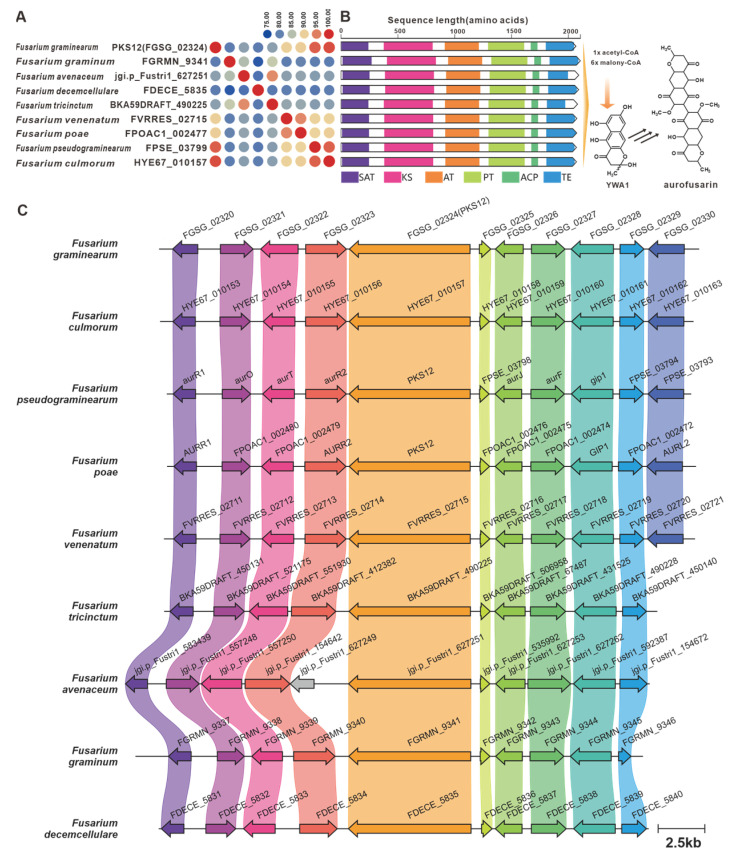
Aurofusarin biosynthesis. (**A**) Comparison of the amino acid sequence identity of PKS12 and its homologues. (**B**) Domain comparison of PKS12 and its homologues: different colored regions represent different domains. (**C**) Comparison of the BGC for aurofusarin and its similar BGCs: homologous genes are connected by a band of the same color.

**Figure 19 jof-09-00850-f019:**
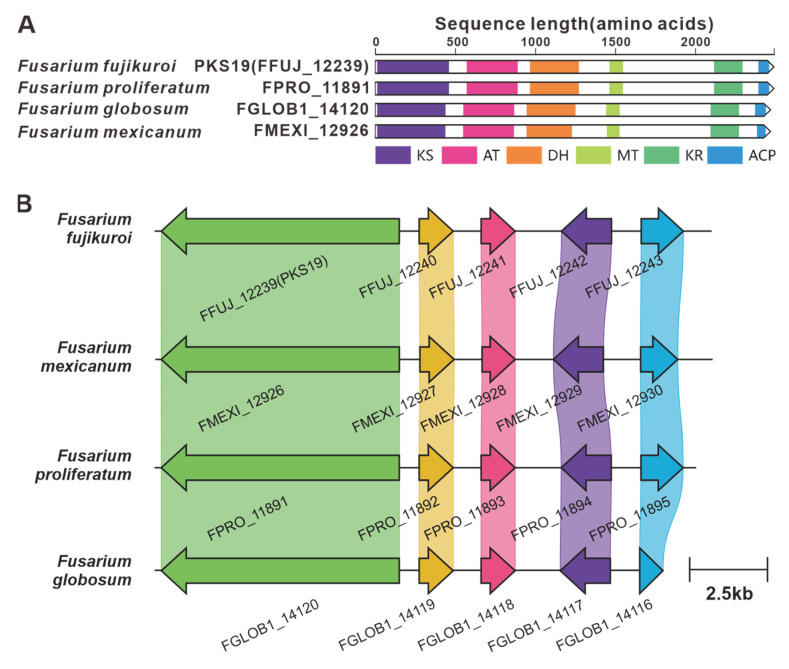
Fujikurin biosynthesis. (**A**) Comparison of the amino acid sequence identity of PKS19 and its homologues: different colored regions represent different domains. (**B**) Comparison of the BGC for fujikurin and its similar BGCs: homologous genes are connected by a band of the same color.

**Figure 20 jof-09-00850-f020:**
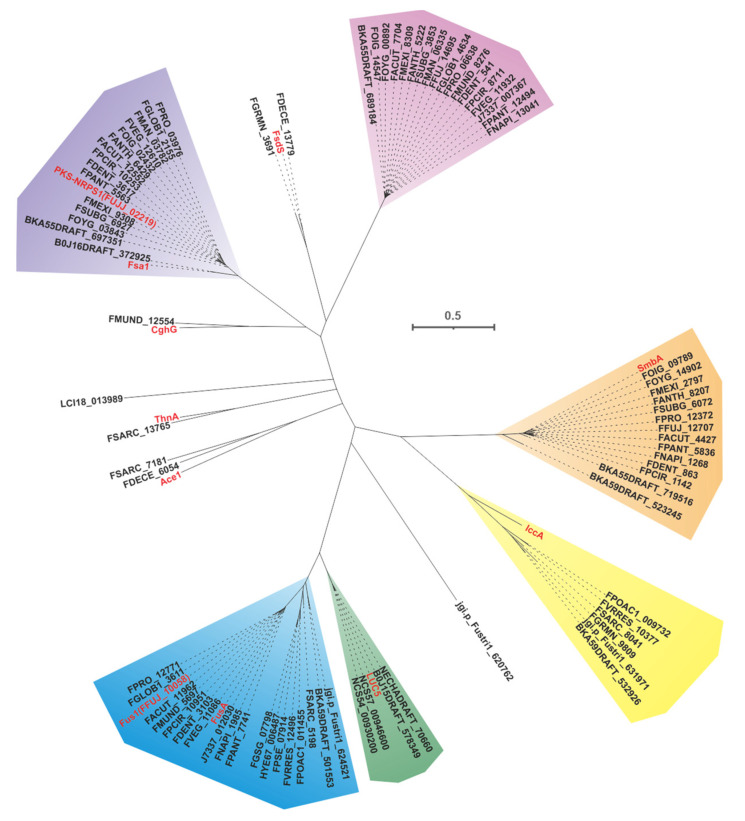
Evolutionary tree-based cluster analysis for PKS-NRPSs. Different colored shaded backgrounds represent different clusters, and red entries represent identified PKS-NRPSs.

**Figure 21 jof-09-00850-f021:**
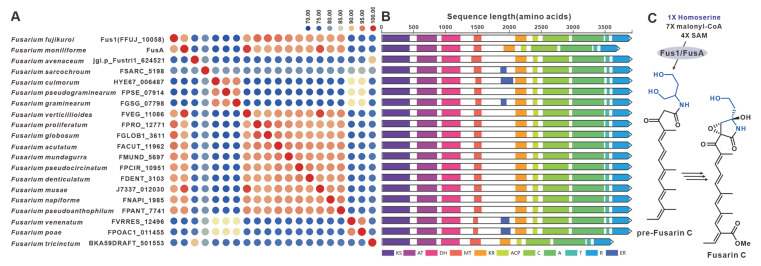
Fusarin C biosynthesis. (**A**) Comparison of the amino acid sequence identity of Fus1, FusA and their homologues. (**B**) Domain comparison of Fus1, FusA and their homologues: homologous genes are connected by a band of the same color. (**C**) The biosynthetic pathway for Fusarin C.

**Figure 22 jof-09-00850-f022:**
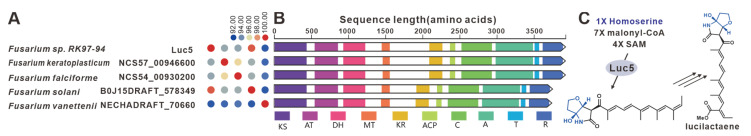
Lucilactaene biosynthesis. (**A**) Comparison of the amino acid sequence identity of Luc5 and its homologues. (**B**) Domain comparison of Luc5 and its homologues: homologous genes are connected by a band of the same color. (**C**) The biosynthetic pathway for lucilactaene.

**Figure 23 jof-09-00850-f023:**
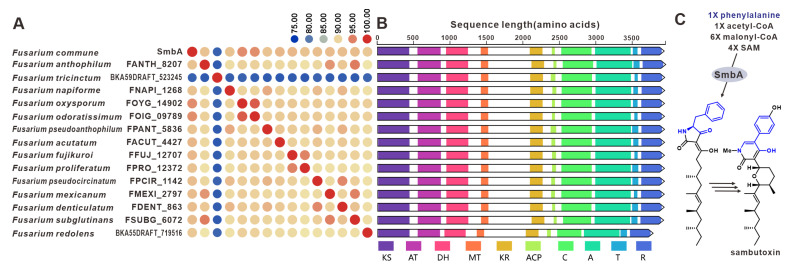
Sambutoxin biosynthesis. (**A**) Comparison of the amino acid sequence identity of SmbA and its homologues: homologous genes are connected by a band of the same color. (**B**) Domain comparison of SmbA and its homologues. (**C**) The biosynthetic pathway for mycotoxin.

**Figure 24 jof-09-00850-f024:**
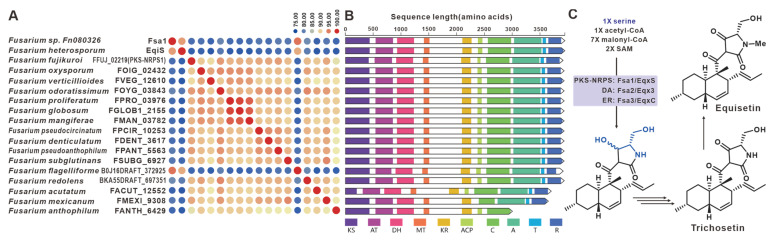
Equisetin compound biosynthesis. (**A**) Comparison of the amino acid sequence identity of Fsa1, EqiS, FFUJ_02219 and their homologues. (**B**) Domain comparison of Fsa1, EqiS, FFUJ_02219 and their homologues: homologous genes are connected by a band of the same color. (**C**) The biosynthetic pathway for equisetin compounds.

**Figure 25 jof-09-00850-f025:**
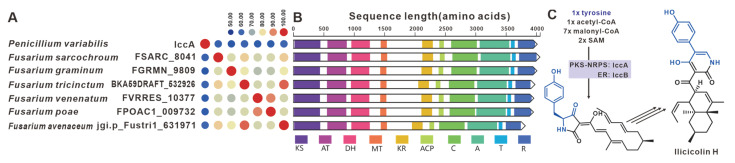
Ilicicolin H compound biosynthesis. (**A**) Comparison of the amino acid sequence identity of IccA and its homologues. (**B**) Domain comparison of IccA and its homologues: homologous genes are connected by a band of the same color. (**C**) The biosynthetic pathway for ilicicolin H.

## Data Availability

Not applicable.

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
