# Peer review of "Bioinformatic Analysis of Secondary Metabolite Biosynthetic Potential in Pathogenic Fusarium"

_jof, 2023, doi:10.3390/jof9080850_

Round 1

Reviewer 1 Report

The authors present a very broad and very detailed genomic analysis of gene clusters and gene encoding enzymes for secondary metabolites in Fusarium species. The analysis will be beneficial for the community. Fusarium is a very important genus for food production, some of the species affect human health directly. Mycotoxins that are part of the presented analysis are very important because they threaten plant health and food safety. For all these reasons I think that the manuscript should be published after revisions.

The manuscript suffers from several disadvantages, some of them are due to the nature of the analysis and therefore cannot be significantly improved but some of them can be improved.

1. The analysis is based on existing tools, there is no use in a sophisticated algorithm or AI to improve and precise the findings.

2. Big parts of the manuscript read as a grocery list, while there is some importance I would try to minimize this part. For example, it is not clear to me why to describe in such detail the sequence similarity between the genes. What is the meaning of it. If the sequence similarity can serve as a readout to learn more about the evolution of the genes this should be further investigated. The authors stop their analysis at the mere description of the similarity.

3. The authors present several phylogenetic trees of gene families as a follow-up to points 1 and 2 I request the authors to compare at least in some cases to the species tree - that may reveal horizontal gene transfer that can be very interesting here.

More editorial comments

4. The style of the title of the figures is not unified. The same is true for the figure legends. In such complicated manuscript I would prefer a title that is informative (conclusion) and not descriptive (put words to the figure). I would prefer a brief description of how the analysis was done - as done in figure 2. I brought some specific examples but this point should be improved throughout.

Fig. 1 more details on the analysis and more informative title

Fig. 2 Explain better what is the difference between the shaded GFC and the one that are not

Fig. 3 More informative title, better description. What are the leaves of the tree that are colorless.

5. Paragraph starting on line 190 (but elsewhere in the manuscript). Reads to me as a “discussion” insertion withing the results section. It is better to separate the discussion of individual genes from the analysis unless they serve to test a hypothesis or have a unique genomic feature that is further investigated.

6. The relationships between sequence similarity and structural similarity (Fig. 5) are not clear enough. Do we see higher similarity in the structure than revealed from the sequence?

I could definitely understand the manuscript. 

Author Response

Q1:The authors present a very broad and very detailed genomic analysis of gene clusters and gene encoding enzymes for secondary metabolites in Fusarium species. The analysis will be beneficial for the community. Fusarium is a very important genus for food production, some of the species affect human health directly. Mycotoxins that are part of the presented analysis are very important because they threaten plant health and food safety. For all these reasons I think that the manuscript should be published after revisions.

A1:We thank the reviewers for their recognition of our work and their praise of its significance.

Q2: 1. The analysis is based on existing tools, there is no use in a sophisticated algorithm or AI to improve and precise the findings.

A2:Yes, this work is based on existing raw information tools. We are very complimentary of the complex algorithms or artificial intelligence tools suggested by the reviewers, which is what we are looking for. We are delving into the application of AI in biosynthesis, such as more efficient tools for core enzyme identification. Unfortunately, our progress is slow. At the moment, we are still at the level of data mining using conventional tools.

Q3: 2. Big parts of the manuscript read as a grocery list, while there is some importance, I would try to minimize this part. For example, it is not clear to me why to describe in such detail the sequence similarity between the genes. What is the meaning of it. If the sequence similarity can serve as a readout to learn more about the evolution of the genes this should be further investigated. The authors stop their analysis at the mere description of the similarity.

A3:The reviewer's comments are accurate and to the point. In the NRPS and PKS chapters, we describe similarity analyses of several groups of core enzymes in terms of their primary sequence and domain composition, as well as the gene clusters in which they are located. On the one hand, these descriptions give a realistic picture of their similarities and their general distribution in the species concerned; on the other hand, these similarity analyses provide clues for functional studies of such genes (gene clusters), such as heterologous expression or gene knockdown. It is even possible to avoid homologues to highly similar and identified enzymes in the search for certain novel enzymes. In short, these studies can provide important clues for biosynthetic research. Of course, we have done our best to revise the manuscript to make the descriptions more readable and to increase the interest of the manuscript while maintaining the scientific quality.

Q4: 3. The authors present several phylogenetic trees of gene families as a follow-up to points 1 and 2 I request the authors to compare at least in some cases to the species tree - that may reveal horizontal gene transfer that can be very interesting here.

A4:We thank the reviewers for pointing out the results of possible horizontal gene transfer of highly homologous gene family members distributed in different pathogenic Fusarium species. We highly agree with this observation. In fact, we had this in mind during the writing of the manuscript, but did not formulate our opinion so clearly. We thank the reviewers for their constructive comments, which have been added to the Discussion section.

Q5: 4. The style of the title of the figures is not unified. The same is true for the figure legends. In such complicated manuscript I would prefer a title that is informative (conclusion) and not descriptive (put words to the figure). I would prefer a brief description of how the analysis was done - as done in figure 2. I brought some specific examples but this point should be improved throughout.

A5:Many thanks to the reviewers for their constructive suggestions, which we believe are important for improving the quality of the manuscript. In response to the title of Figure 4, a conclusive description is indeed needed. In addition, the titles of the remaining 24 figures were double-checked and corrected one by one for informative and scientific purposes.

Q6:Fig. 1 more details on the analysis and more informative title

A6:Based on your constructive suggestions, we have made the following changes after careful consideration.

 “Distribution of six different types of gene clusters in 35 pathogenic Fusarium species. (A) Evolutionary relationships of the 35 pathogenic Fusarium species were constructed based on orthologous single-copy genes. (B) The BGC types were counted to visualize based on AntiSMASH predictions.”

Q7:  Fig. 2 Explain better what is the difference between the shaded GFC and the one that are not

A7:Thank you for your careful consideration and we are sorry for this oversight. The members of the shaded GCF are very similar and are therefore defined as XXX_GCF. The composition of the unshaded GCF is more complex, and only the known BGCs in it are labelled; it is also described in the manuscript, and we have included the description in the title here.

Q8:  Fig. 3 More informative title, better description. What are the leaves of the tree that are colorless.

A8:Thank you again for your careful review. The leaves of the tree that are colorless indicate sesquiterpene sequences from 35 Fusarium and which could not be identified in this work. Considering the brevity of the title information, we have not specified this. In fact, the first sentence of the title, "Cluster analysis of 209 sesquiterpene sequences." already covers this information.

Q9: 5. Paragraph starting on line 190 (but elsewhere in the manuscript). Reads to me as a “discussion” insertion withing the results section. It is better to separate the discussion of individual genes from the analysis unless they serve to test a hypothesis or have a unique genomic feature that is further investigated.

 A9:We thank the reviewers for their different interpretations of the paragraph (starting with line 190). This paragraph is a description of the results of the clustering analysis (Figure 3) of the evolutionary tree-based clustering of sesquiterpene cyclases and is therefore the results section. In this paragraph and in the following results, it is possible that our presentation was mixed with some descriptions of a discursive nature, which led to different interpretations by the reviewers. We have done our best to revise it to avoid inappropriate interpretations of the relevant statements.

Q10: 6. The relationships between sequence similarity and structural similarity (Fig. 5) are not clear enough. Do we see higher similarity in the structure than revealed from the sequence?

A10:Thanks to the reviewers for their attention to detail. Sequence similarity refers to the similarity of the amino acid sequences that make up the primary structure of a protein, i.e. the amino acid sequences are compared. Here, structural similarity refers to the spatial alignment of the three-dimensional structure based on alpha-fold simulations implemented by PyMOL. Sesterterpenoid synthases are chimeric enzymes, i.e. they have two structural domains. Structural domains with the same function are essentially identical in their overall structure, although the primary sequences that make up these structural domains can be quite different. Therefore, when compared spatially, they appear to be similar, but in fact their primary sequences are not very similar.

Comments on the Quality of English Language

Q11: I could definitely understand the manuscript. 

A11: Thanks to the reviewers' acknowledgement of the English writing, the manuscript was again carefully read and corrected in an attempt to make its presentation more accurate and authentic.

Reviewer 2 Report

Extensive work has been carried out to evaluate the BGCs of the Fusarium species and analysing new sesterterpene synthases and PKS_NRPS clusters. This manuscript contributes to the scientific community that is interested in Fusarium, the fungal species that is considered one of the major mycotoxins producers. Therefore, I accept the manuscript for publication.

Minor corrections

- Revise the formatting of references 

Author Response

Response to Reviewer 2

Q1: Extensive work has been carried out to evaluate the BGCs of the Fusarium species and analysing new sesterterpene synthases and PKS_NRPS clusters. This manuscript contributes to the scientific community that is interested in Fusarium, the fungal species that is considered one of the major mycotoxins producers. Therefore, I accept the manuscript for publication.

A1: We thank the reviewers for recognizing this work.

Minor corrections

Q2: Revise the formatting of references 

A2: The references section does contain sporadic detail joining errors, and we have reviewed the literature individually and made corrections. For example, the redundant DOI No. in line 939 has been removed, and the species name in line 1118 has been italicized.

Reviewer 3 Report

Excellent work! I have nothing to object to regarding the protocols used. I would have liked some results to be accompanied by a significance level, but I was running some of the programs mentioned and saw that they were not in the output.

Gilchrist, C.L.M.; Chooi, Y.-H. clinker & clustermap.js: automatic generation of gene cluster comparison figures. 921

BIOINFORMATICS 2021, 37, 2473-2475.

 …ption of over LC18013491(Figure 10B).  (add space)

nrps9  vs NRPS9  (keep the font)

KS-AT-DH-MT-KR-ACP-C(Figure S56). To better understand their 713 (add space) 

The structural similarity between Fusarin C catalysed by FusA or Fus1(Figure 2) and 746 (add space) 

732 number of NRPSs was the largest, while the number of hybird enzymes was the least. Hybrid?

 391for the biosynthesis of the hybird compound fusaristatin A in F. graminearum [71]. Hybrid?

Author Response

Response to Reviewer 3

Q1: Excellent work! I have nothing to object to regarding the protocols used. I would have liked some results to be accompanied by a significance level, but I was running some of the programs mentioned and saw that they were not in the output.

A1: We thank the reviewers for their high praise. The fact that some runs did not produce the expected results may be due to a lack of detail in the description of our parameters. In the revised manuscript we have tried our best to present the relevant parameters clearly.

Comments on the Quality of English Language

Q2: Gilchrist, C.L.M.; Chooi, Y.-H. clinker & clustermap.js: automatic generation of gene cluster comparison figures. 921 BIOINFORMATICS 2021, 37, 2473-2475.

Q3:  …ption of over LC18013491(Figure 10B).  (add space)

Q4: KS-AT-DH-MT-KR-ACP-C(Figure S56). To better understand their 713 (add space) 

Q5: The structural similarity between Fusarin C catalysed by FusA or Fus1(Figure 2) and 746 (add space) 

A2-A5: We thank the reviewers for their careful scrutiny and the relevant missing spaces have been added.

Q6: nrps9 vs NRPS9 (keep the font)

A6: Thanks to the carefulness of the reviewers, the deficiencies in the misuse of nrps9 and NRPS9 have been corrected.

Q7: 732 number of NRPSs was the largest, while the number of hybird enzymes was the least. Hybrid?

Q8: 391for the biosynthesis of the hybird compound fusaristatin A in F. graminearum [71]. Hybrid?

A7-A8:Sorry for this spelling error, I have checked and corrected the word elsewhere in the manuscript.

Round 2

Reviewer 1 Report

The reviewers hardly changed the manuscript. Some not adequate change in the figure legends. The manuscript still reads as a grocery list and the authors did nothing to improve it. The authors did not do even one comparison between the species trees and the gene trees. I would expect to do at least 10 comparisons or provide evidence for horizontal gene transfer.

/

Author Response

The reviewers hardly changed the manuscript. Some not adequate change in the figure legends. The manuscript still reads as a grocery list and the authors did nothing to improve it. The authors did not do even one comparison between the species trees and the gene trees. I would expect to do at least 10 comparisons or provide evidence for horizontal gene transfer.

This is the second round of comments from reviewer 1. First of all, we would like to express our sincere thanks to reviewer 1 for his two careful reviews, which we believe will help to improve the quality of the manuscript. We will now respond to the second round of reviews in detail.

Q1:The reviewers hardly changed the manuscript.

A1:During the first round of revision, we valued the comments of the three reviewers and did our best to make careful revisions based on the comments. On this basis, the language of the manuscript was revised by a native English speaker. The revised manuscript is labelled "jof-2530492R1-marked.pdf" and has been uploaded in the first round of the system.

Q2: Some not adequate change in the figure legends.

A2: Following the advice of reviewer 1 in the first round, we have rewritten the figure captions for figures 1 and 2. Informative titles have been given to Figures 4 and 6, and the figure captions for Figures 10, 12-13, 15-19, 21-25 have been revised in detail to make them more scientific and rigorous.

In this round, we have examined each of the 25 figure legends carefully, and revised some bugs and modified the legends of Figure 7 in detail, which will make them more informative.

Q3: The manuscript still reads as a grocery list and the authors did nothing to improve it.

A3: Thanks to reviewer 1 for this seemingly unkind comment. In fact, as you can see, this is a biosynthesis-type manuscript for biosynthesis mining of the Fusarium genome using the existing Fusarium biosynthetic information as a clue. A large number of details of Fusarium (toxin) biosynthetic analyses are the main part of the manuscript. The presentation of these detailed results is regularly organized according to the type of biosynthetic core genes, i.e. terpenes, NRPS, PKS, PKS-NRPS in order from 3.2 to 3.5. In the discussion section, such core genes are summarised separately. In the conclusion section, the results of these analyses are also summarised.

This type of manuscript organisation is quite common. It may be that the manuscript is too long and too detailed to give the reviewer a bad impression of a " grocery list". The relevant parts of our manuscript were carefully re-read and revised to ensure the rigour and scientific quality of the manuscript.

Q4: The authors did not do even one comparison between the species trees and the gene trees.

A4: Species trees were constructed to analyse the distribution of gene clusters for species specificity. The interpretation of the species tree is presented in the first half of the first paragraph of result 3.1, described before Figure 1A.

The results presented in Figures 3, 7, 20 and Figures S2, S4, S7, S27 of the supplementary material were obtained by evolutionary tree analysis, but these results are also cluster analyses. All 3.2-3.5 of the Results section were obtained by analyzing the results of these cluster analyses according to their specific types.

Q5: I would expect to do at least 10 comparisons or provide evidence for horizontal gene transfer.

A5: We thank reviewer 1 for suggesting horizontal gene transfer in the previous round and for the refined requirements in this round. We believe these suggestions are very beneficial to the quality of the manuscript. With the guidance of reviewer 1, we researched the literatures and found 20 papers on horizontal gene transfer, many of which involved about Fusarium metabolites biosynthesis. Some of the literatures pointed out that the biosynthesis of Trichothecene, fumonisin, in Fusarium are related to horizontal gene transfer. The relevant literatures have been cited in the manuscript.

  1. Proctor, R.H.; Hao, G.; Kim, H.S.; Whitaker, B.K.; Laraba, I.; Vaughan, M.M.; McCormick, S.P. A Novel Trichothecene Toxin Phenotype Associated with Horizontal Gene Transfer and a Change in Gene Function in Fusarium. Toxins (Basel) 2022, 15, doi:10.3390/toxins15010012.
  2. Sieber, C.M.; Lee, W.; Wong, P.; Munsterkotter, M.; Mewes, H.W.; Schmeitzl, C.; Varga, E.; Berthiller, F.; Adam, G.; Guldener, U. The Fusarium graminearum genome reveals more secondary metabolite gene clusters and hints of horizontal gene transfer. PLoS One 2014, 9, e110311, doi: 10.1371/journal.pone.0110311.
  3. Proctor, R.H.; Van Hove, F.; Susca, A.; Stea, G.; Busman, M.; van der Lee, T.; Waalwijk, C.; Moretti, A.; Ward, T.J. Birth, death and horizontal transfer of the fumonisin biosynthetic gene cluster during the evolutionary diversification of Fusarium. Mol Microbiol 2013, 90, 290-306, doi:10.1111/mmi.12362.
  4. Liu, S.; Wu, B.; Lv, S.; Shen, Z.; Li, R.; Yi, G.; Li, C.; Guo, X. Genetic Diversity in FUB Genes of Fusarium oxysporum Suggests Horizontal Gene Transfer. Front Plant Sci 2019, 10, 1069, doi: 10.3389/fpls.2019.01069.
  5. Mehrabi, R.; Bahkali, A.H.; Abd-Elsalam, K.A.; Moslem, M.; Ben M'barek, S.; Gohari, A.M.; Jashni, M.K.; Stergiopoulos, I.; Kema, G.H.; de Wit, P.J. Horizontal gene and chromosome transfer in plant pathogenic fungi affecting host range. FEMS Microbiol Rev 2011, 35, 542-554, doi:10.1111/j.1574-6976.2010.00263.x.
  6. Walton, J.D. Horizontal gene transfer and the evolution of secondary metabolite gene clusters in fungi: an hypothesis. Fungal Genet Biol 2000, 30, 167-171, doi:10.1006/fgbi.2000.1224.
  7. Wang, H.; Sun, S.; Ge, W.; Zhao, L.; Hou, B.; Wang, K.; Lyu, Z.; Chen, L.; Xu, S.; Guo, J.; et al. Horizontal gene transfer of Fhb7 from fungus underlies Fusarium head blight resistance in wheat. Science 2020, 368, doi: 10.1126/science.aba5435.
  8. Nielsen, S.M.; de Gier, C.; Dimopoulou, C.; Gupta, V.; Hansen, L.H.; Norskov-Lauritsen, N. The capsule biosynthesis locus of Haemophilus influenzae shows conspicuous similarity to the corresponding locus in Haemophilus sputorum and may have been recruited from this species by horizontal gene transfer. Microbiology 2015, 161, 1182-1188, doi:10.1099/mic.0.000081.
  9. Chen, J.Y.; Liu, C.; Gui, Y.J.; Si, K.W.; Zhang, D.D.; Wang, J.; Short, D.P.G.; Huang, J.Q.; Li, N.Y.; Liang, Y.; et al. Comparative genomics reveals cotton-specific virulence factors in flexible genomic regions in Verticillium dahliae and evidence of horizontal gene transfer from Fusarium. New Phytol 2018, 217, 756-770, doi:10.1111/nph.14861.
  10. Li, Q.; Yang, L.; Xiang, D.; Wan, Y.; Wu, Q.; Huang, W.; Zhao, G. The complete mitochondrial genomes of two model ectomycorrhizal fungi (Laccaria): features, intron dynamics and phylogenetic implications. Int J Biol Macromol 2020, 145, 974-984, doi: 10.1016/j.ijbiomac.2019.09.188.
  11. van Dam, P.; Rep, M. The Distribution of Miniature Impala Elements and SIX Genes in the Fusarium Genus is Suggestive of Horizontal Gene Transfer. J Mol Evol 2017, 85, 14-25, doi:10.1007/s00239-017-9801-0.
  12. Simbaqueba, J.; Catanzariti, A.M.; Gonzalez, C.; Jones, D.A. Evidence for horizontal gene transfer and separation of effector recognition from effector function revealed by analysis of effector genes shared between cape gooseberry- and tomato-infecting formae speciales of Fusarium oxysporum. Mol Plant Pathol 2018, 19, 2302-2318, doi:10.1111/mpp.12700.
  13. Michael, A.J. Evolution of biosynthetic diversity. Biochem J 2017, 474, 2277-2299, doi:10.1042/BCJ20160823.
  14. Vlaardingerbroek, I.; Beerens, B.; Rose, L.; Fokkens, L.; Cornelissen, B.J.; Rep, M. Exchange of core chromosomes and horizontal transfer of lineage-specific chromosomes in Fusarium oxysporum. Environ Microbiol 2016, 18, 3702-3713, doi:10.1111/1462-2920.13281.
  15. Shoguchi, E. Gene clusters for biosynthesis of mycosporine-like amino acids in dinoflagellate nuclear genomes: Possible recent horizontal gene transfer between species of Symbiodiniaceae (Dinophyceae). J Phycol 2022, 58, 1-11, doi:10.1111/jpy.13219.
  16. Khanppnavar, B.; Chatterjee, R.; Choudhury, G.B.; Datta, S. Genome-wide survey and crystallographic analysis suggests a role for both horizontal gene transfer and duplication in pantothenate biosynthesis pathways. Biochim Biophys Acta Gen Subj 2019, 1863, 1547-1559, doi: 10.1016/j.bbagen.2019.05.017.
  17. Gao, S.; Gold, S.E.; Wisecaver, J.H.; Zhang, Y.; Guo, L.; Ma, L.J.; Rokas, A.; Glenn, A.E. Genome-wide analysis of Fusarium verticillioides reveals inter-kingdom contribution of horizontal gene transfer to the expansion of metabolism. Fungal Genet Biol 2019, 128, 60-73, doi: 10.1016/j.fgb.2019.04.002.
  18. Kalia, V.C.; Lal, S.; Cheema, S. Insight in to the phylogeny of polyhydroxyalkanoate biosynthesis: horizontal gene transfer. Gene 2007, 389, 19-26, doi: 10.1016/j.gene.2006.09.010.
  19. Czislowski, E.; Fraser-Smith, S.; Zander, M.; O'Neill, W.T.; Meldrum, R.A.; Tran-Nguyen, L.T.T.; Batley, J.; Aitken, E.A.B. Investigation of the diversity of effector genes in the banana pathogen, Fusarium oxysporum, reveals evidence of horizontal gene transfer. Mol Plant Pathol 2018, 19, 1155-1171, doi:10.1111/mpp.12594.
  20. Deng, M.R.; Guo, J.; Li, X.; Zhu, C.H.; Zhu, H.H. Granaticins and their biosynthetic gene cluster from Streptomyces vietnamensis: evidence of horizontal gene transfer. Antonie Van Leeuwenhoek 2011, 100, 607-617, doi:10.1007/s10482-011-9615-9.